# Surprising phenotypic diversity of cancer-associated mutations of Gly 34 in the histone H3 tail

Brandon R Lowe[1], Rajesh K Yadav[1†], Ryan A Henry[2‡], Patrick Schreiner[3], Atsushi Matsuda[4,5], Alfonso G Fernandez[1], David Finkelstein[3], Margaret Campbell[1], Satish Kallappagoudar[1], Carolyn M Jablonowski[1], Andrew J Andrews[2], Yasushi Hiraoka[4,5], Janet F Partridge[1]*

[1]Department of Pathology, St. Jude Children's Research Hospital, Memphis, United States; [2]Department of Cancer Biology, Fox Chase Cancer Center, Philadelphia, United States; [3]Department of Bioinformatics, St. Jude Children's Research Hospital, Memphis, United States; [4]Advanced ICT Research Institute Kobe, National Institute of Information and Communications Technology, Kobe, Japan; [5]Graduate School of Frontier Biosciences, Osaka University, Suita, Japan

*For correspondence:
janet.partridge@stjude.org

Present address: [†]Department of Biochemistry, All India Institute of Medical Sciences, Patna, India; [‡]Department of Chemistry and Biochemistry, Wilkes University, Wilkes-Barre, United States

Competing interests: The authors declare that no competing interests exist.

**Abstract** Sequencing of cancer genomes has identified recurrent somatic mutations in histones, termed oncohistones, which are frequently poorly understood. Previously we showed that fission yeast expressing only the H3.3G34R mutant identified in aggressive pediatric glioma had reduced H3K36 trimethylation and acetylation, increased genomic instability and replicative stress, and defective homology-dependent DNA damage repair. Here we show that surprisingly distinct phenotypes result from G34V (also in glioma) and G34W (giant cell tumors of bone) mutations, differentially affecting H3K36 modifications, subtelomeric silencing, genomic stability; sensitivity to irradiation, alkylating agents, and hydroxyurea; and influencing DNA repair. In cancer, only 1 of 30 alleles encoding H3 is mutated. Whilst co-expression of wild-type H3 rescues most G34 mutant phenotypes, G34R causes dominant hydroxyurea sensitivity, homologous recombination defects, and dominant subtelomeric silencing. Together, these studies demonstrate the complexity associated with different substitutions at even a single residue in H3 and highlight the utility of genetically tractable systems for their analysis.

## Introduction

The fundamental regulation of DNA within eukaryotes is coordinated through highly conserved histone proteins that package DNA into the nucleus. Proteins that regulate post-translational modifications of histone proteins or of DNA, or that control nucleosome assembly, disassembly, or movement, are frequently targeted in cancer (*Huether et al., 2014*; *Roy et al., 2014*; *Shen and Laird, 2013*). More recently, mutations within the histone genes themselves have been identified in disease (*Behjati et al., 2013*; *Lu et al., 2016*; *Nacev et al., 2019*; *Schwartzentruber et al., 2012*; *Tessadori et al., 2017*; *Wu et al., 2012*). These histone mutations arise predominantly in one of two genes encoding the histone variant protein H3.3. H3.3 is expressed throughout the cell cycle and, in contrast to the replication-dependent histone H3.1 and H3.2, is deposited into chromatin outside of S-phase (reviewed in *Kallappagoudar et al., 2015*). Oncogenic H3.3 mutations frequently alter residues that are key sites of post-translational modification (K27M or K36M), or mutate G34, which affects the methylation of the neighboring K36 (H3K36me [*Lewis et al., 2013*]). The K27M and K36M mutants, found in 84% of diffuse intrinsic pontine gliomas and 95% chondroblastomas, respectively (*Behjati et al., 2013*; *Schwartzentruber et al., 2012*; *Wu et al., 2012*), exert dominant effects

by inhibition of the relevant methyltransferase complexes through binding the mutant histone tails, reducing methylation of total cellular histone H3 pools at K27 or K36, respectively (*Brown et al., 2014*; *Fang et al., 2016*; *Jain et al., 2019*; *Lewis et al., 2013*; *Lu et al., 2016*; *Zhang et al., 2017*). We know much less, however, about the role of histone mutants that lack obvious dominant effects on post-translational modification of the remaining histone pools (*Bjerke et al., 2013*; *Lewis et al., 2013*; *Nacev et al., 2019*; *Tessadori et al., 2017*). For example, although H4 K91 mutants (Q and R) impact acetylation and ubiquitination of the mutant H4 tail, and H3.3 G34R and V decrease K36me3 on the mutant H3.3 tail, no alterations in post-translational modification have been reported on the wild-type H4 or H3 proteins expressed in these cells (*Lewis et al., 2013*; *Tessadori et al., 2017*). However, pronounced redistribution of the K36me3 mark was seen in a pediatric glioblastoma cell line derived from a H3.3 G34V patient (KNS42), correlating with a switch to transcriptional activation of genes normally expressed in early embryonic development (*Bjerke et al., 2013*). Whether this change is caused by the histone mutation, is a downstream effect of transcriptional change, or the result of another mutation frequently found in H3.3 G34V mutant glioma, for example, mutation of p53 or H3.3 chaperone and remodeler proteins DAXX and ATRX (*Bjerke et al., 2013*; *Korshunov et al., 2016*; *Schwartzentruber et al., 2012*; *Wu et al., 2012*), is currently unclear.

We set out to address the role of mutations in histone H3, one of the most highly conserved proteins in eukaryotes, using the highly tractable fission yeast. This system excels for evaluation of the biological effects of mutations divorced from confounding effects of additional mutations in tumors and can be stripped of the complexity of additional wild-type histone H3 variants within cells. We previously reported that fission yeast manipulated to express only G34R mutant histone H3 (H3-G34R) showed defective H3K36 acetylation (H3K36ac) and reduced levels of H3K36me3 (*Yadav et al., 2017*). These chromatin changes correlated with some transcriptional changes, including upregulation of antisense transcripts, enhanced subtelomeric silencing, as well as defects in replication and homologous recombination (HR), which we hypothesized were causal of genomic instability in these cells. However, it remained unclear if particular chromatin changes could be linked to specific defects.

Here, we investigate other mutants of H3G34. H3-G34V mutation is less frequent than G34R in pediatric high-grade glioma, but the tumors occur in the same cortical location in patients of similar age (*Schwartzentruber et al., 2012*; *Wu et al., 2012*). Based on these findings, we expected similar behavior of H3-G34V and H3-G34R mutants. Surprisingly, although H3K36me3 was reduced in both mutants, other phenotypes were quite distinct, with G34V showing no defect in H3K36 acetylation, no chromosome loss, no replicative stress, and no defects in HR, but enhanced sensitivity to γ-irradiation (IR). We queried whether these differences were due to the size or the charge of the substitution. By generating a panel of H3-G34 mutant yeast, we found that G34K, which like G34R incorporates a basic charge, also showed reduced K36me3 and chromosome loss but G34W, a bulky uncharged substitution found in H3.3 in nearly all giant cell tumors of bone (*Behjati et al., 2013*), did not alter H3K36me3 or genome stability and did not sensitize cells to the DNA-damaging agents tested. We further examined which effects were dominant, and found that only some, including the HU sensitivity and HR defects in H3-G34R mutants, and silencing of subtelomeric domains in both H3-G34R and G34K mutants was evident in cells expressing a mixture of wild-type and mutant H3. In contrast, chromosome segregation was corrected and most DNA damage sensitivities were lost when cells additionally express wild-type H3. This study reveals the pivotal role of H3 Gly 34 in orchestrating many cellular responses, some that apparently act on a local level and others that are able to exert dominant effects.

## Results

Fission yeast have three genes that code for a single histone H3 protein (*Figure 1A*; *Matsumoto and Yanagida, 1985*). Strains were derived that express only one H3 and one H4 gene (*hht2⁺* and *hhf2⁺*) (*Mellone et al., 2003*), which maintain histone protein levels similar to those in wild-type strains (*Yadav et al., 2017*). Previously, we introduced a mutation into *hht2⁺* (*Figure 1A*) to generate strains that express only the G34R mutant form of histone H3 (*Yadav et al., 2017*). In the current study, we derived a panel of strains that express only H3-G34 mutants, and compared

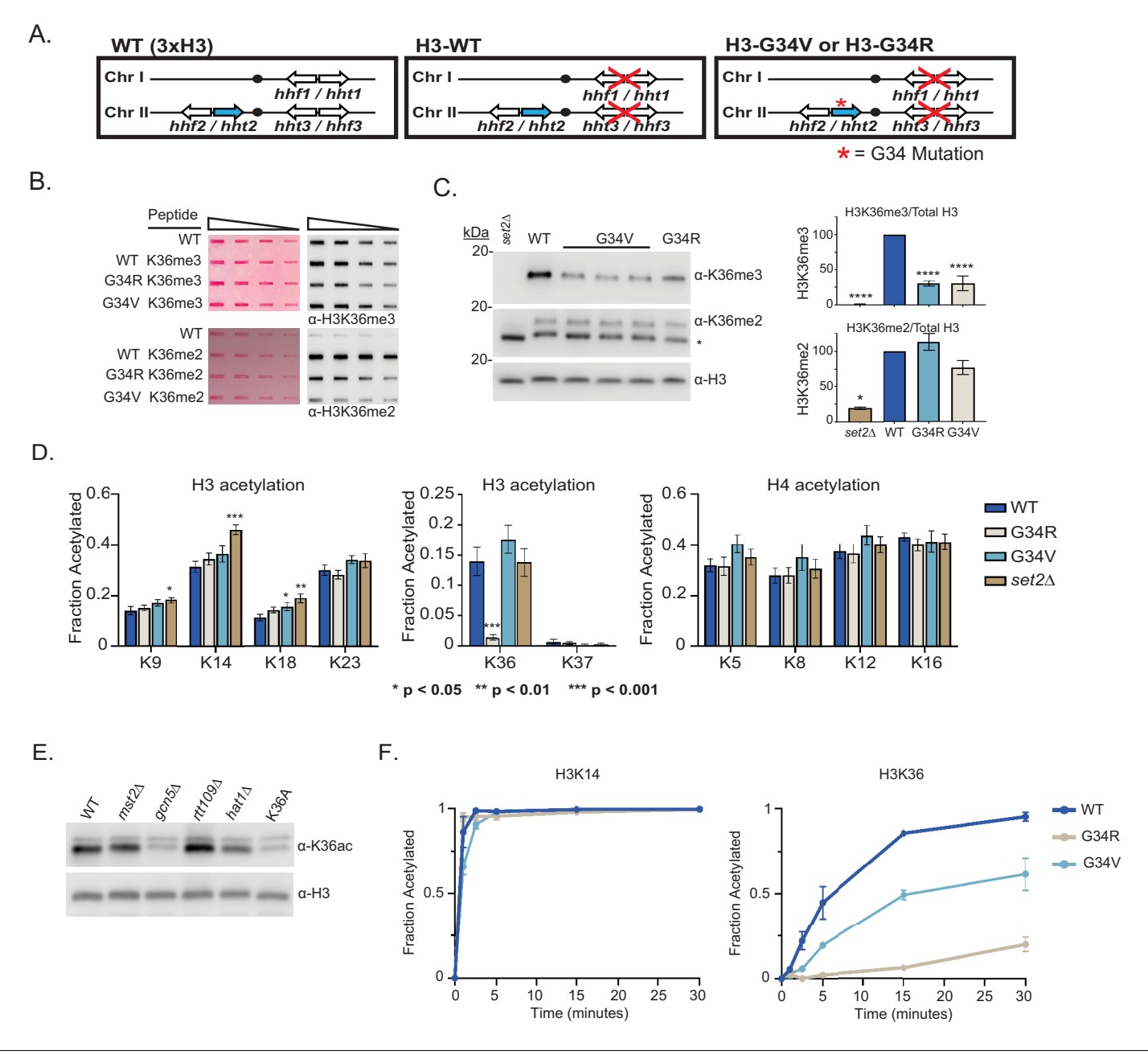

**Figure 1.** Differential modification of H3K36 in H3-G34R and H3-G34V mutants. (**A**) Scheme of the histone H3 (*hht*) and histone H4 (*hhf*) genes in *Schizosaccharomyces pombe* (*S. pombe*) highlighting the H3 gene (*hht2*) in which mutations were engineered (blue). (**B**) Dot blot analysis to quantitatively assess recognition of WT, G34R, and G34V peptides bearing K36 di- or tri-methyl modifications by anti-K36 methyl antibodies. Ponceau-stained blots were used as the loading control (left). (**C**) Western blot analysis of K36me2, K36me3, and total H3 in H3-WT, H3-G34V, H3-G34R, and *set2Δ* chromatin extracts. Star marks non-specific band. (Right) quantification of K36 methylation relative to total H3 (K36me3: three replicates for H3-WT, *set2Δ*, and H3-G34R and eight replicates for H3-G34V; K36me2: two replicates for H3-WT, H3-G34R, and *set2Δ* and five replicates for H3-G34V). For K36me3 blot, **** represents significant difference p<0.0001 from H3-WT strain. (**D**) Mass spectrometry-based quantification of acetylation of specific lysines in histone H4 and H3 tails in histones purified from H3-WT, H3-G34R, H3-G34V, and *set2Δ* strains (nine biological replicates for H3-WT, H3-G34R, and *set2Δ* data and six biological replicates for H3-G34V). H3K27ac analysis was excluded as highly variable. (**E**) Western blot analysis of H3K36ac in lysates of (3xH3) WT, and acetyltransferase mutants *mst2Δ*, *gcn5Δ*, *hat1Δ*, and *rtt109Δ* with sole copy H3-K36A negative control (anti-H3K36ac Abnova PAB31320), and total H3 as loading control. (**F**) In vitro histone acetylation assay using recombinant Gcn5 and recombinant WT, G34R, or G34V H3 and monitoring H3K14ac and H3K36ac. Data from each time point represents the mean ± SEM from three biological replicates. The online version of this article includes the following figure supplement(s) for figure 1:

**Figure supplement 1.** Differential modification of H3K36 in H3-G34R and H3-G34V mutants.

them with H3-G34R strains. 'Single copy' histone H3 and H4 strains were used throughout and are named **H3-WT** and **H3-G34X** in the text, and '**WT**' and '**G34X**' in the figures. Additional mutations were also generated in the single H3 background unless denoted (**3xH3**) representing strains with three copies of H3/H4 (*Supplementary file 1*).

## H3K36me3 is reduced in H3-G34V mutants

In fission yeast, all stages of H3 K36 methylation (mono, di, and tri) are carried out by a single enzyme, Set2 (*Morris et al., 2005*). Experiments monitoring H3K36 methylation by mass spectrometry in human cells have shown that expression of G34R or V mutant H3.3 reduces H3K36me2 and me3 on the same histone tail (*Lewis et al., 2013*). Since H3-G34R mutant fission yeast exhibit a pronounced reduction in H3K36me3 (*Yadav et al., 2017*), we asked whether H3K36me was similarly affected by H3-G34V. We first characterized antibodies for recognition of H3K36 methylation in peptides containing a H3G34V mutant tail (*Figure 1B*, *Supplementary file 2*). Using an anti-K36me3 antibody that bound efficiently to G34V mutant tails, western analyses showed a marked reduction in H3K36me3 in chromatin extracts from H3-G34V compared with H3-WT cells (*Figure 1C*). In contrast, H3K36me2 appeared unchanged on western analysis of H3-G34V compared with H3-WT strains, but is upregulated in H3-G34V cells since antibody recognition of K36me2 is notably reduced on the G34V mutant H3 tail. A similar effect on K36me2 was seen in G34R cells (*Yadav et al., 2017*). Thus H3K36me3 is reduced and H3K36me2 accumulates in both H3-G34V and H3-G34R cells. Since Set2 is the sole K36 methyltransferase in fission yeast, this result suggests that there may be a defect in Set2-mediated trimethylation of K36me2 G34V/R templates, or that there is heightened activity of a K36me3 demethylase. To probe these possibilities, we performed ChIP analysis of Set2 in G34V/R cells, but saw no change in Set2 association with chromatin at the sites tested (*Figure 1—figure supplement 1A*). Second, we asked whether loss of Epe1, a proposed H3 demethylase (*Trewick et al., 2007*), influences levels of H3K36 methylation, but we saw no consistent change in K36me2 or me3 in *epe1Δ* strains (*Figure 1—figure supplement 1B*).

## H3-G34R but not H3-G34V cells exhibit differences in H3 acetylation

Since H3-G34R mutation results in a marked reduction in H3K36ac (*Yadav et al., 2017*), we asked whether K36ac was also affected by G34V mutation. Targeted quantitative mass spectrometry of histone H3 and H4 acetylation was performed on histones extracted from H3-WT, H3-G34V, H3-G34R, and *set2Δ* cells (*Figure 1D*; *Supplementary file 3*; *Kuo and Andrews, 2013*). We found that H3K36ac was greatly reduced in H3-G34R, but was unaffected in H3-G34V cells. In conclusion, H3-G34R, but not G34V, appears to inhibit efficient acetylation of H3 K36, whereas both mutants show loss of H3K36me3.

## Gcn5 acetylates H3K36, but cannot acetylate K36 on H3-G34R tail in vitro

To determine which histone acetyltransferase mediates H3K36ac, we assessed H3K36ac levels in cells lacking various histone acetyltransferases. Gcn5 has previously been implicated in H3K36ac (*Pai et al., 2014*) and consistent with this, using antibody that was largely specific to K36ac (*Figure 1—figure supplement 1C*), we found that *gcn5Δ* cells have low H3K36ac levels, similar to H3-K36A cells which lack H3K36ac (*Figure 1E*). To extend this analysis, since GCN5 has many targets, including H3K14, H3K18, and H3K23, we asked if the specific loss of K36ac on H3-G34R mutant H3 could be recapitulated in vitro. Using recombinant fission yeast Gcn5 with recombinant WT or G34 mutant histone H3s, we used mass spectroscopy to monitor acetylation at individual sites in the H3 tail over time (*Kuo and Andrews, 2013*; *Kuo et al., 2014*). Whereas Gcn5 acetylated most sites on WT, G34R, and G34V mutant H3 templates equivalently, acetylation of K36 on G34R H3 was strongly reduced and was slowed down on G34V H3 (*Figure 1F* and *Figure 1—figure supplement 1D*). Together, the in vivo and in vitro data suggest that the G34R mutation specifically reduces acetylation on H3 K36, while acetylation at other sites remains largely unaffected. These data are consistent with structural studies of Gcn5—substrate complexes, which predicted that residues just N-terminal to the reactive lysine would modulate substrate affinity, and that replacement of the small residue (glycine or alanine) at position −2 (in this case H3G34 relative to H3K36 target) with a larger residue would likely impact Gcn5 activity (*Figure 1—figure supplement 1E*; *Poux and Marmorstein, 2003*).

## H3-G34V mutants are not sensitive to replication stress

Altered H3K36 post-translational modification is associated with defective repair of DNA damage (*Jha and Strahl, 2014*; *Li et al., 2013*; *Pai et al., 2014*; *Pfister et al., 2014*). Consistent with this, H3-G34R cells with reduced H3K36ac and H3K36me3 are sensitive to replicative stress (*Yadav et al., 2017*). To query the role of H3K36ac in replicative stress, we plated fivefold serial dilutions of G34V and G34R cells onto media containing drugs, and assessed cell growth after several days (*Figure 2A*). H3-G34R and *set2Δ* cells were sensitive to chronic exposure to hydroxyurea (HU, a ribonucleotide reductase inhibitor that depletes dNTPs), whereas H3-G34V cells were not. G34V cells were also not sensitive to DNA alkylation (methyl methanesulfonate, MMS), whereas both G34R and *set2Δ* cells were sensitive. Finally, H3-G34V cells were unaffected by exposure to the DNA topoisomerase I ligase inhibitor CPT, whereas H3-G34R and *set2Δ* showed slight or pronounced sensitivity, respectively. At the concentrations used, these genotoxins predominantly affect DNA replication or require DNA replication to inflict damage. Thus while H3-G34R and *set2Δ* are sensitive to several forms of replicative stress, H3-G34V cells are not. Since both H3-G34V and H3-G34R have reduced K36me3, the sensitivity of H3-G34R cells to replicative stress cannot be attributed to a reduction in H3K36me3 but may be due to loss of H3K36ac.

## H3-G34V, but not H3-G34R mutants are sensitive to γ-IR and IR mimetics

Cells lacking *set2⁺* are sensitive to γ-IR and IR mimetics (*Pai et al., 2014*) but H3-G34R cells are not (*Yadav et al., 2017*). We tested H3-G34V cells for sensitivity to γ-IR by transiently irradiating H3-WT, H3-G34R, H3-G34V, and *set2Δ* cells, and comparing viability by plating single cells and scoring colony formation after several days of growth (*Figure 2B*). Notably, H3-G34V cells were sensitive to γ-IR, whereas as seen before, H3-G34R were not. H3-G34V and *set2Δ* cells were also sensitive to chronic exposure to the IR mimetics, bleomycin and zeocin, whereas H3-G34R were not (*Figure 2C*). Thus the H3-G34V but not H3-G34R mutation renders cells sensitive to both γ-IR and chronic exposure to IR mimetics.

## H3-G34V mutants are not defective for DNA break repair by homologous recombination

We next tested HR-mediated double-strand (ds) break repair efficiency in H3-G34V cells, since H3-G34R and *set2Δ* mutants are defective in HR-mediated DNA repair (*Yadav et al., 2017*). *leu1-32* mutant cells were transformed with a fragment of wild-type *leu1⁺* and *leu1⁺* transformants that arose by HR were scored (see *Figure 2—figure supplement 1A*). In comparison to *set2Δ* and H3-G34R cells, which showed significant reduction in HR activity, the H3-G34V mutant showed no defect in HR (*Figure 2—figure supplement 1B*). However, this assay probes HR efficiency at a single site, so we next performed a genetic epistasis analysis to ask more generally whether H3-G34V cells were competent for HR. To this end, we asked whether the DNA damage sensitivity of H3-G34V cells is enhanced on deletion of the major HR repair protein Rad51, which would suggest that HR is proficient in H3-G34V cells. Indeed we found that combination of *rad51Δ* and H3-G34V mutation enhanced DNA damage sensitivity, whereas combination of *rad51Δ* with H3-G34R rendered cells no more sensitive than *rad51Δ* (*Figure 2D*). Together these assays indicate that H3-G34V cells rely on HR for efficient resolution of DNA damage, whereas H3-G34R cells are defective for HR. Note that we could not use epistasis analyses to probe other types of DNA damage repair in H3-G34V mutants since H3-G34V cells are only sensitive to IR or IR mimetics, and mutants defective in non-homologous end joining (NHEJ), such as *ku70Δ* and *lig4Δ*, are not sensitive to these insults (*Manolis et al., 2001*).

## H3-G34R but not H3-G34V cells exhibit genomic instability

H3-G34R cells exhibit genomic instability (*Yadav et al., 2017*). Since H3-G34V cells exhibit sensitivity to γ-IR, we asked if genomic stability was also compromised in H3-G34V cells. We initially monitored the loss of a non-essential minichromosome Ch16 (*Niwa et al., 1989*) from H3-G34V cells. As expected, cells lacking the heterochromatin protein Swi6[HP1] displayed high frequencies of Ch16 loss (8.5%, *Figure 2E*; *Allshire et al., 1995*). H3-G34R cells also lose the minichromosome at a

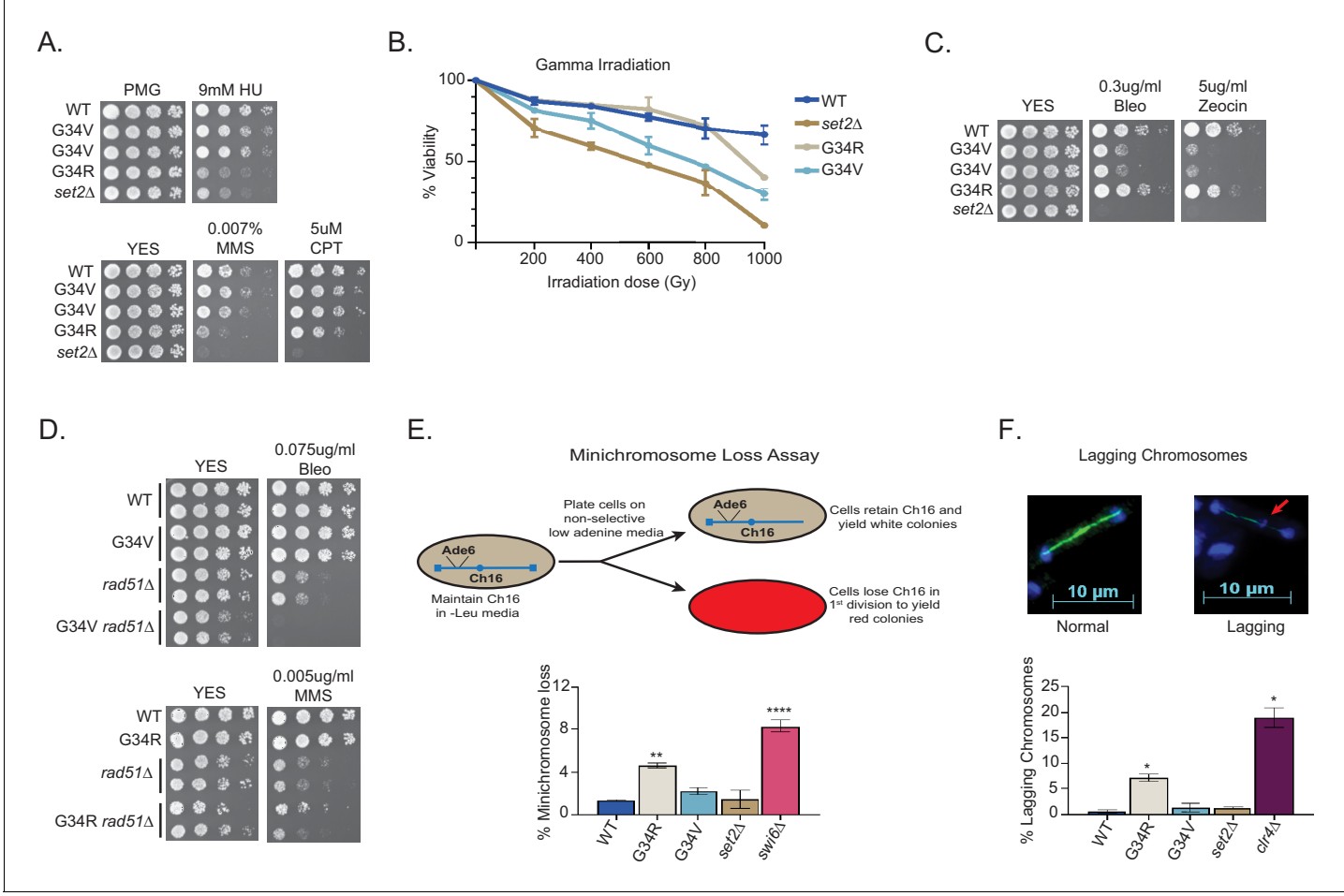

**Figure 2.** DNA damage sensitivity and chromosomal stability differ between H3-G34R and H3-G34V mutants. (**A**) Fivefold serial dilutions showing the effect of hydroxyurea (HU; a ribonucleotide reductase inhibitor that depletes dNTPs), methyl methanesulfonate (MMS; a DNA-alkylating agent), and camptothecin (CPT; blocks topoisomerase one ligase activity) on growth of the indicated strains. (**B**) Effect of γ−irradiation (IR) exposure on viability of H3-WT, H3-G34R, H3-G34V, *set2Δ*, and H3-G34V *set2Δ* cells. Data represent mean ± SEM from two independent experiments using four biological replicates. (**C**) Serial dilution assay showing the effect of bleomycin and zeocin, two IR mimetics, on the indicated strains. (**D**) Serial dilution assay assessing epistasis of mutants with HR pathway. Bleomycin sensitivity of H3-WT, H3-G34V, *rad51Δ*, and H3-G34V *rad51Δ* cells (top) and MMS sensitivity of H3-WT, H3-G34R, *rad51Δ*, and H3-G34R *rad51Δ* cells (bottom). (**E**) Frequency of cells that lose the non-essential minichromosome Ch16 in H3-WT, H3-G34V, H3-G34R, *set2Δ*, and *swi6Δ* cells. Mean ± SEM from four to eight biological replicates is shown. ** denotes a significant difference of p<0.01 and ****p<0.0001 compared with the H3-WT strain. (**F**) Example of a normal anaphase and one with a lagging chromosome (red arrow) (top). Frequency of late anaphase cells with a lagging chromosome in H3-WT, G34R, G34V, *set2Δ*, and *clr4Δ* (bottom). Mean ± SEM from four to eight biological replicates. * represents significant difference of p<0.05 with the H3-WT strain.

The online version of this article includes the following figure supplement(s) for figure 2:

**Figure supplement 1.** G34V exhibits no defect in HR.

significantly elevated frequency (4.3%) (*Yadav et al., 2017*), whereas H3-G34V and *set2Δ* cells display low levels of chromosome loss (2.2% and 1.8%), similar to H3-WT cells (1.2%).

As an alternate assay to measure chromosome segregation defects, we quantified chromosomal DNA segregation in late anaphase cells (with a spindle length >10 microns) (*Figure 2F*; *Ekwall et al., 1995*). Since fission yeast have just three chromosomes, defects in chromosome segregation (lagging chromosomes) can be scored by monitoring the presence of DAPI-stained material at sites other than the ends of the spindle in late anaphase cells. We found that 7.4% of H3-G34R cells exhibited chromosome mis-segregation. In contrast, G34V and *set2Δ* cells showed no or little chromosome mis-segregation, respectively (0.5%, 1%), compared to wild-type cells (0.5%) and *clr4Δ*

cells that lack heterochromatin (17.5%). We conclude from these assays that the genome of H3-G34V cells is relatively stable when compared with H3-G34R cells.

## Generation of a panel of H3-G34 mutant strains and analysis of H3K36me state

The surprising differences in the behavior of H3-G34V and H3-G34R mutants prompted us to ask whether H3-G34R phenotypes are caused by the large size or the basic charge of the arginine substitution, since valine is, like glycine, small and uncharged. To address this question, we generated a panel of sole copy H3-G34 mutant strains, with G34 replaced by the bulky uncharged residue tryptophan (W), the negatively charged glutamine (Q), the positively charged lysine (K), and methionine (M) (*Figure 3A*). Using the approaches outlined previously to determine whether antibodies against H3K36 methylation can still bind mutant H3 tails (*Figure 3B*), we then assessed levels of H3K36me2 and me3 by western analysis on chromatin extracted from the H3-G34 mutant strains, and plotted quantification from western blots relative to H3-WT (*Figure 3C*). This analysis showed that H3K36me3 is specifically reduced in H3-G34K (as well as H3-G34R and H3-G34V), but is not reduced in H3-G34W, M, or Q mutants. Furthermore, H3K36me2 is elevated in all H3-G34 mutant strains compared with H3-WT, when the reduced binding of anti-H3K36me2 antibodies to G34 mutant tails is taken into account.

## H3-G34 mutants show a range of DNA damage sensitivities and genomic stability

H3-G34K, V, and R mutants showed loss of K36me3 and gain of K36me2, whereas H3-G34M, W, and Q mutants retained H3K36me3 and gained K36me2. To ask the functional consequence of these changes, we assessed the sensitivity of the different mutant strains to DNA damage. We found that of all the H3-G34 mutants, only H3-G34R was sensitive to hydroxyurea. In contrast, both strains that bear the basic substitutions (H3-G34R and H3-G34K) were sensitive to the alkylating agent MMS and to topoisomerase inhibition CPT. We found that MMS sensitivity was not altered in strains bearing the bulky H3-G34W, G34M, or the negatively charged G34Q substitutions (*Figure 3D*). This would suggest that alkylation sensitivity is not linked to the size of the substitution at H3G34. On testing the γ-IR mimetics zeocin and bleomycin, we found that H3-G34K cells, like H3-G34V mutants showed sensitivity, whereas H3-G34M, G34W, G34Q, and G34R cells did not (*Figure 3E*). We also monitored cell viability following γ-IR and found that H3-G34K and H3-G34V but not H3-G34R strains were sensitive to γ-IR, and that H3-G34K was more sensitive than either *set2Δ* or H3-G34V mutant cells (*Figure 3F*). These experiments demonstrate the surprising complexity of DNA damage sensitivities of the different H3-G34 mutants, with some mutants (H3-G34V and H3-G34R) showing completely non-overlapping sensitivities, and others (H3-G34K and H3-G34R, and H3-G34K and H3-G34V) showing overlap for sensitivity to alkylating agents and IR, respectively. Since all three mutants H3-G34R, H3-G34V, and H3-G34K have reduced K36me3 and enhanced DNA damage sensitivity, K36me3 levels likely serve a critical role in safeguarding the genome, but the sensitivity of the different mutants to specific DNA-damaging insults must be influenced by additional factors.

## H3-G34R and G34K exhibit chromosome loss but only H3-G34R appears defective for HR-mediated DNA repair

Since H3-G34K mutants were sensitive to DNA damage, we tested whether they showed genomic instability in the absence of exogenous DNA damage. Monitoring chromosome missegregation during anaphase, we found that H3-G34K cells had elevated levels of chromosome missegregation (7.4%), which were reproducibly higher than seen in H3-G34R cells (5.9%) (*Figure 3G*). In contrast, none of the other H3-G34 mutants (V, M, W, or Q) showed a significant chromosome loss compared with H3-WT (0.7%).

H3-G34R cells are defective for HR-mediated DNA repair, and we previously proposed that chromosome missegregation in H3-G34R mutants might be due to the persistence of DNA repair intermediates in mitotic cells (*Yadav et al., 2017*). We therefore asked whether the other mutant exhibiting chromosome loss, H3-G34K, was similarly defective for HR-mediated repair. By introducing the *rad51* mutant allele and monitoring epistasis between H3-G34K and *rad51Δ*, we found that double mutant H3-G34K *rad51Δ* cells were more sensitive to both MMS treatment and bleomycin

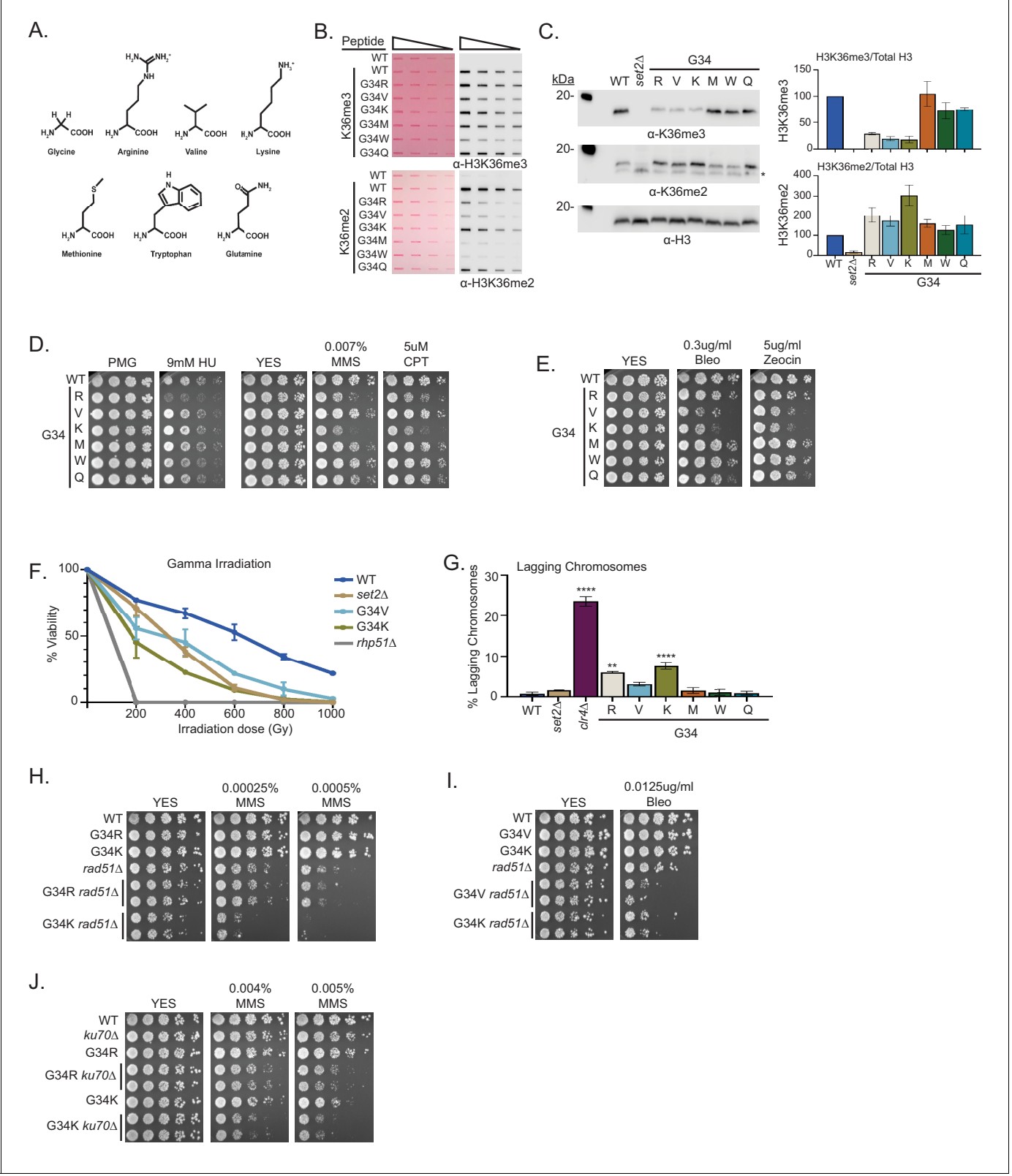

**Figure 3.** A panel of mutants at H3-G34 exhibit distinct effects on H3K36 methylation and different DNA-damage sensitivities. (**A**) Structure of glycine and amino acid substitutions used in experiments. (**B**) Dot blot analysis to quantitatively assess recognition of WT, H3-G34R, V, K, M, W, or Q peptides bearing K36me2 or K36me3 modifications by anti-K36 methyl antibodies. Peptides were loaded in twofold serial dilutions and Ponceau staining was used as the loading control (left). (**C**) Western blot analysis of H3K36me3, K36me2, and total H3 in *set2Δ*, H3-WT, H3-G34R, V, K, M, W, and Q

*Figure 3 continued on next page*

**Figure 3 continued**

chromatin-fractionated cellular extracts (left). The * symbol represents a non-specific band in the H3K36me2 western. Quantification of K36 methylation levels relative to total H3 were calculated from two biological replicates (right). (D) Serial dilution yeast growth assay showing the effect of hydroxyurea (HU), methyl methanesulfonate (MMS), and camptothecin (CPT) on the indicated strains. (E) Serial dilution growth assay showing the effect of bleomycin and zeocin, two irradiation (IR) mimetics, on the indicated strains. (F) Effect of γ-IR exposure on viability of H3-WT, H3-G34R, H3-G34V, H3-G34K, *set2Δ*, and *rad51Δ* cells. Data represent mean ± SEM from eight biological replicates. (G) IF analysis of lagging chromosomes in the indicated strains from three independent experiments. % lagging chromosomes represents the percentage of lagging chromosomes in anaphase cells counted. Over 200 anaphase cells were counted for each strain. ** represents significant difference of p<0.001 and **** a significant difference of p<0.0001 with H3-WT strain. (H) Serial dilution growth assay testing epistasis of H3G34R and H3G34K with HR pathway mutant *rad51Δ* cells. (I) Serial dilution growth assay testing epistasis of H3-G34V with *rad51Δ* HR-deficient cells. (J) Serial dilution growth assay testing epistasis of H3-G34R and H3-G34K with *ku70Δ* NHEJ-deficient cells.

treatment than either single mutant alone (*Figure 3H and I*), suggesting that H3-G34K is competent for HR. Similar results were found for H3-G34V (*Figures 2D* and *3I*). In contrast, H3-G34R *rad51Δ* double mutants were no more sensitive to MMS treatment than *rad51Δ* alone (*Figure 3H*), supporting that H3-G34R cells are defective in HR.

We also tested whether H3-G34K and H3-G34R mutants were deficient in NHEJ. H3-G34R or H3-G34K double mutants with *ku70Δ* showed elevated sensitivity to MMS compared with the single mutants alone (*Figure 3J*), suggestive that NHEJ is intact in both the H3-G34R and H3-G34K strains. Unfortunately we could not perform this test for H3-G34V strains as they do not share genotoxin sensitivity with mutants defective for NHEJ. Together, these results suggest that HR is deficient only in H3-G34R mutants, and that NHEJ is intact in both H3-G34R and H3-G34K cells. However, we note that in the absence of NHEJ repair assays, our conclusions about NHEJ proficiency based solely on drug sensitivity are indirect.

## H3-G34V, H3-G34R, and H3G34K mutants show distinct transcriptional profiles

To try to rationalize the distinct phenotypes of H3-G34R, H3-G34V, and H3-G34K mutants, we hypothesized that the mutations result in distinct transcriptional regulation. We performed RNA-seq in triplicate to compare transcriptional profiles of H3-G34R, H3-G34K, H3-G34V, *set2Δ*, and H3-WT cells within the same experiment (*Supplementary file 4*). Quantification of spike-in controls demonstrated that there was no global deregulation of transcripts in the mutant backgrounds, consistent with our previous data for H3-G34R and *set2Δ* mutants (*Yadav et al., 2017*). In total, 325 genes are upregulated, and 153 downregulated in *set2Δ* (employing cutoff ± 1.5-fold, FDR 5%), similar to the results we obtained previously (*Yadav et al., 2017*) and as initially reported in (3xH3) *set2Δ* strains (*Matsuda et al., 2015*; *Suzuki et al., 2016*). H3-G34R, H3-G34K, and H3-G34V exhibited fewer changes in gene expression, with 78, 128, and 94 genes increased in expression, and 82, 95, and 89 reduced, respectively (*Supplementary file 4*). Comparison of genes that were differentially regulated between H3-G34R, H3-G34K, and H3-G34V showed little overlap, with only 44 genes misregulated in all three strains (*Supplementary file 4*).

Previously, our chromosome-wide analysis of transcripts had revealed that genes that lie within the sub-subtelomeric regions of chromosomes I and II (herein designated as 120 Kb regions at the ends of chromosomes I and II) called ST domains (*Buchanan et al., 2009*; *Matsuda et al., 2015*; *Yadav et al., 2017*) were repressed in H3-G34R cells, whereas these domains are upregulated in *set2Δ* cells (*Yadav et al., 2017*). In this study, we found that the transcriptional profile of ST regions in H3-G34V cells more closely resembles that of H3-WT cells than the repression observed in H3-G34R, or the increased expression in *set2Δ* cells (*Figure 4A,B*). In contrast, H3-G34K showed a slight trend toward transcriptional repression of ST regions (in particular on the left end of chromosome II). Further analysis of gene sets that were differentially regulated revealed that 16 of the 31 genes that were downregulated in both G34R and G34K strains were located within ST domains of Chr I or II, in contrast to just two of the 29 genes upregulated in both strains (*Supplementary file 4*). We further confirmed the repression of ST domain gene expression in both H3-G34R and H3-G34K cells by RT-PCR analysis of *fah1*[+] and *grt1*[+] transcripts compared with controls (*Figure 4C*).

Intriguingly, the histone acetyltranferases Gcn5 and Mst2 also negatively regulate silencing of the ST domains at the termini of chromosomes I and II (*Gómez et al., 2005*; *Nugent et al., 2010*;

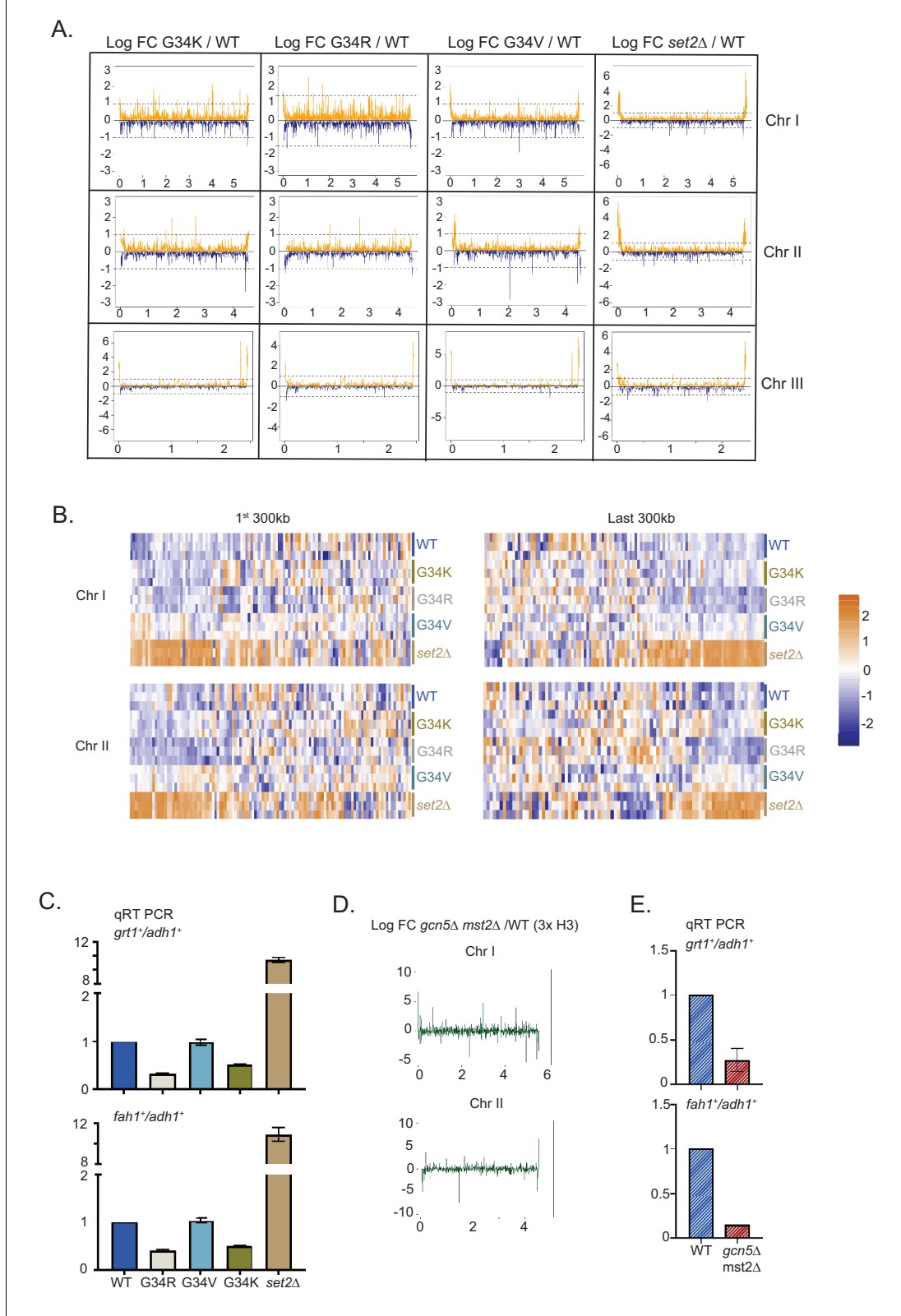

**Figure 4.** Distinct transcriptional outcomes for H3-G34 mutants: substitution with basic residues suppressing some subtelomeric transcripts, as seen in strains deficient in H3 K36 acetylation. (**A**) RNA-seq profiles for chromosomes I, II, and III comparing Logfold change ratios for H3-G34K/H3-WT, H3-G34V/H3-WT, H3-G34R/H3-WT, or *set2Δ*/H3-WT plotted against chromosome coordinates. (**B**) Zoomed-in regions of Chr I (first 300 Kb and last 300 Kb, top) and Chr II (first 300 Kb and last 300 Kb, bottom) showing Z scores of log2 CPM for individual biological replicates. (**C**) qRT-PCR validation of ST

*Figure 4 continued on next page*

*Figure 4 continued*

genes *fah1*⁺ and *grt1*⁺ expression relative to *adh1*⁺ expression from two independent biological replicates. Samples were normalized to the WT-H3 strain. Subtelomeric transcripts in H3-G34R and H3-G34K are repressed compared with H3-WT and upregulated in *set2Δ*. (D) Chromosome-wide plots of transcriptional regulation in *gcn5Δmst2Δ* cells (3xH3) compared with wild type for Chr I and Chr II. Data reanalyzed from *Nugent et al., 2010*. (E) qRT-PCR validation of *fah1*⁺ and *grt1*⁺ expression relative to *adh1*⁺ expression from two independent biological replicates. Samples were normalized to the WT-H3 strain. Subtelomeric transcripts in *gcn5Δmst2Δ* cells (3xH3) are reduced compared with wild type.

*Figure 4D*, data re-analyzed from *Nugent et al., 2010*), and by RT-PCR analysis of *fah1*⁺ and *grt1*⁺ expression, we confirmed downregulation of ST gene expression in *gcn5Δmst2Δ* cells (*Figure 4E*). Thus the downregulation of subtelomeric regions in G34R cells in which K36 acetylation is reduced is mirrored in cells lacking H3 K36 acetyltransferase function.

## H3-G34R cells and cells lacking H3K36ac accumulate Sgo2 at subtelomeres and have increased cytological 'knobs'

Subtelomeric domains are surprisingly the most highly condensed regions of fission yeast chromatin and are microscopically visible as DAPI (4',6-diamidino-2-phenylindole)-stained 'knobs' (*Matsuda et al., 2015*). Knob formation correlates with transcriptional silencing within subtelomeric domains, and requires Set2 H3K36 methyltransferase and the Shugoshin protein (Sgo2; *Matsuda et al., 2015*; *Tashiro et al., 2016*). Sgo2 is an important component of the tension-sensing machinery that assembles at centromeres during mitosis and relays turn-off of the mitotic checkpoint once kinetochores have properly attached to spindle microtubules (*Watanabe, 2005*). Recently however Sgo2 has been found at subtelomeres in interphase cells, where it promotes silencing, and regulates timing of replication of ST domains (*Tashiro et al., 2016*). Since H3-G34R but not H3-G34V shows extensive subtelomeric gene silencing, we examined 'knob' formation in H3-WT, H3-G34R, H3-G34V, and *set2Δ* cells. The levels of visibly condensed chromatin were similar in both single copy and (3xH3) wild-type cells. Notably, the proportion of cells with visible knobs and their number per cell was significantly elevated in H3-G34R but not in H3-G34V cells (*Figure 5A*), and as expected, knobs were reduced in cells lacking Set2 (*Matsuda et al., 2015*). A previous study showed that H3-K36Q mutant cells that mimic H3K36 acetylation and block K36 methylation dramatically loose knobs (*Matsuda et al., 2015*). Since Gcn5 acetylates H3K36, and sub-telomeric domains are silenced in cells lacking Mst2 and Gcn5, we predicted that knobs would be increased in this mutant background. Indeed knob formation in *gcn5Δmst2Δ* mutants is elevated, similar to levels in the K36ac-deficient H3-G34R cells (*Figure 5B*).

Since knob formation is dependent on Sgo2 (*Tashiro et al., 2016*), we hypothesized that Sgo2 may accumulate at ST domains in H3-G34R but not H3-G34V cells, and may also be enriched at ST in cells with reduced H3K36ac. Indeed ChIP revealed Sgo2-FLAG to be significantly enriched at ST domains in H3-G34R cells compared with H3-G34V or H3-WT (*Figure 5C*), and that Sgo2 is highly enriched at ST domains in *mst2Δgcn5Δ* relative to WT controls (*Figure 5D*). Since Sgo2 normally accumulates at centromeres in mitosis and is recruited to ST in interphase (*Kawashima et al., 2010*; *Matsuda et al., 2015*; *Tashiro et al., 2016*), we asked whether H3-G34R influences the cell cycle dependence of its localization. Importantly, Sgo2 accumulation at centromeres in mitosis was unaffected by H3-G34R mutation (*Figure 5E*), but even in mitotically arrested cells, Sgo2 was preferentially enriched at ST domains in H3-G34R cells compared with H3-WT (*Figure 5F*), suggestive that in H3-G34R cells, cell cycle regulation of Sgo2 accumulation at ST is perturbed. In summary, our data suggest that Sgo2 accumulates at subtelomeres in H3-G34R or H3K36ac-deficient cells, correlating with enhanced silencing of ST domains and knob formation in these cells. Furthermore, Sgo2 accumulates at ST domains in mitotic H3-G34R as well as interphase cells, although levels of Sgo2 at centromeres in mitotic H3-G34R cells are similar to H3-WT.

We also performed Sgo2 ChIP in H3-G34K cells. Consistent with enhanced Sgo2 recruitment to silenced ST domains seen in H3K36ac acetyltransferase defective and H3-G34R strains, H3-G34K cells also had elevated Sgo2 at ST sites (*Figure 5G*). However, whether Sgo2 accumulation at ST correlates with reduced H3K36ac is unclear. Mass spectroscopic studies of K36 acetylation are not feasible on H3G34K because of the added complexity in data analysis from introduction of an additional lysine residue in the H3 tail, and western blot analysis is non-informative since antibodies against H3K36ac cannot effectively detect K36ac on G34 mutant H3 tails (*Figure 5—figure supplement 1*).

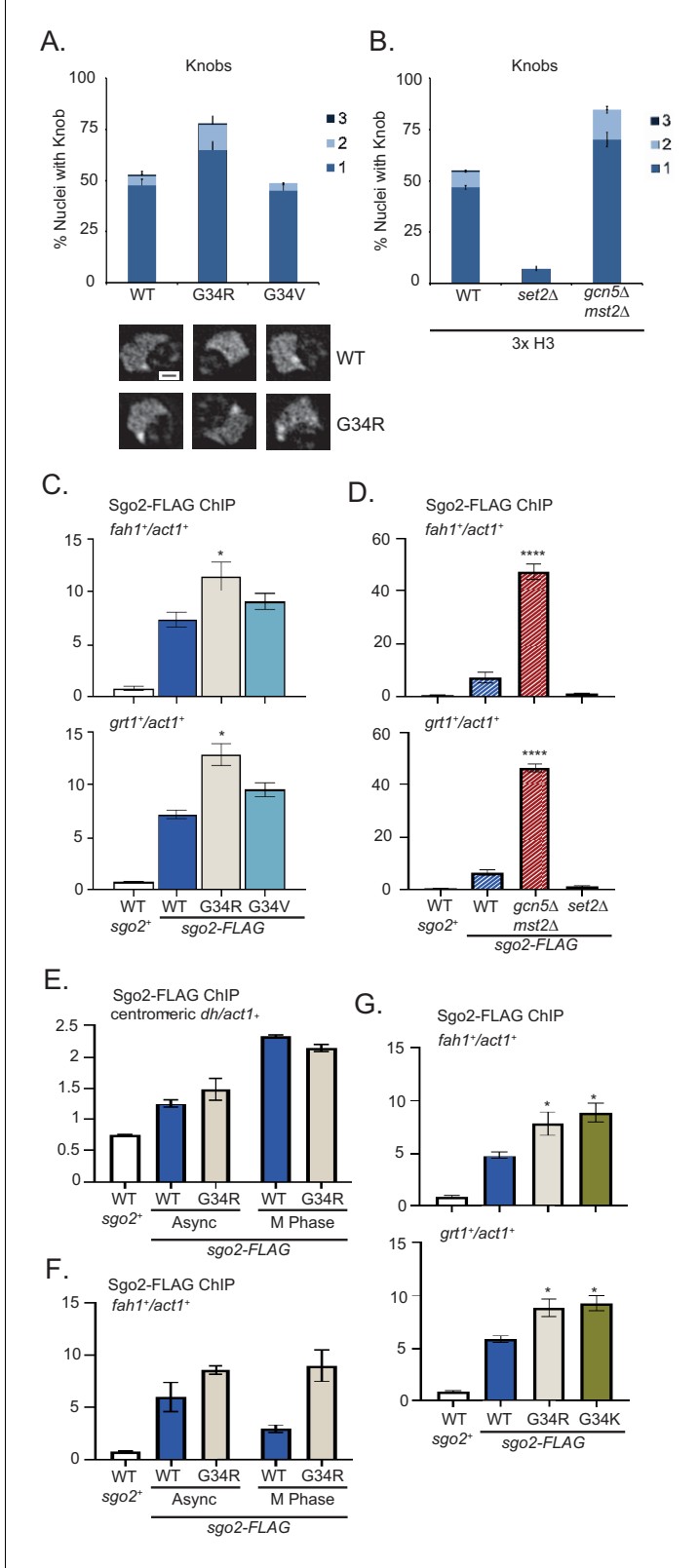

**Figure 5.** Formation of knobs of highly condensed chromatin is enhanced in G34R and K36 acetyltransferase mutants, correlating with enhanced recruitment of Shugoshin to repressed subtelomeric domains. (**A**) Frequency of subtelomeric knob formation observed in H3-WT, H3-G34R, and H3-G34V cells. The number of knobs in a nucleus is shown. Data plots represent mean ± SEM from three independent biological replicates, counting ~200

*Figure 5 continued on next page*

*Figure 5 continued*

total nuclei per strain (upper). Example of nuclear knobs observed in WT and H3-G34; scale bar indicates 0.5 µm (bottom). (B) Frequency of subtelomeric knob formation observed in WT, *set2Δ*, and *gcn5Δmst2Δ* cells (all 3xH3) with the number of knobs in a nucleus shown. Data plots represent mean ± SEM from three independent biological replicates, counting ~200 total nuclei per strain. (C–F) ChIP analysis of Sgo2-FLAG association with the subtelomeric *fah1⁺* and *grt1⁺* loci in (C) H3-WT, H3-G34R, and H3-G34V cells normalized to *act1⁺*. Data were collected and plotted as the mean of six biological replicates ± SEM. * represents a significance of p<0.05 compared with the H3-WT strain. (D) Sgo2-FLAG ChIP in (3xH3) WT, *set2Δ*, and *gcn5Δmst2Δ* cells normalized to *act1⁺*. Data are the mean of four biological replicates ± SEM. **** represents a significance of p<0.0001 compared with the WT strain. (E–F) Sgo2-FLAG ChIP at the centromeric *dh* sequences (E) or subtelomeric *fah1⁺* gene (F) in asynchronous or mitotically arrested *nda3* mutant cells with sole copy H3-WT or H3-G34R (six replicates), and (G) Sgo2-FLAG ChIP in H3-G34K cells normalized to *act1⁺*. Data represents the mean of six biological replicates ± SEM. * represents a significance of p<0.05 compared with the H3-WT strain.

The online version of this article includes the following figure supplement(s) for figure 5:

**Figure supplement 1.** Determining the effect of G34 substitution on antibody recognition of K36ac.

## Select phenotypes of G34R and G34K mutants show dominant effects

An intriguing feature of mutant histone oncoproteins is how they exert their effects when expressed in cells alongside higher levels of wild-type H3 proteins. To determine which, if any, of the H3-G34 mutant phenotypes identified in our studies act dominantly, we generated 'mixed H3' strains that express an H3-G34 mutant gene (*hht3.2*) and two wild-type genes (*hht3.1⁺, hht3.3⁺*) from their endogenous loci (*Figure 1A*), and performed comparative analyses with 3xH3 wild-type strains.

We first asked if H3K36me3 was reduced in mixed H3 strains and found no change in H3K36me3 in mixed copy H3-G34R, H3-G34V, or H3-G34K strains compared with 3xH3 WT (*Figure 6A*). Similarly, we found no dominant increase in levels of H3K36me2 when the mutants were expressed in mixed H3 backgrounds (*Figure 6A*). These results are consistent with the lack of dominant effect of G34R or G34V mutations on K36 methylation in mammalian cells (*Lewis et al., 2013*). Next we assessed H3 K36 acetylation. Due to the sequence difference at G34, we could distinguish K36ac peptides from both G34R and WT H3 proteins extracted from the mixed copy H3-G34R strain. We found that loss of H3K36ac was restricted to the G34R mutant H3 protein, whereas K36ac was present at comparable levels on wild-type H3 purified from the mixed copy G34R and wild-type 3xH3 strains (*Figure 6B*).

To ask whether DNA damage phenotypes were dominant, we assessed growth of 3xH3 strains on chronic exposure to the genotoxins MMS, CPT, and HU. Mixed copy strains grew as well as 3xH3 WT strains, with the notable exception of H3-G34R (3xH3), which like the sole copy H3-G34R mutant, grew less well in the presence of HU (*Figure 6C*, *Figure 6—figure supplement 1A*). We also tested sensitivity to IR mimetics, and found that H3-G34K and H3-G34V mixed copy strains grew as well as WT (3xH3) strains on bleomycin and zeocin (*Figure 6D*).

Our finding of the dominance of HU sensitivity of H3-G34R mixed copy strains led us to ask whether the H3-G34R mutation dominantly affected chromosome segregation or HR-mediated DNA repair. In contrast to the chromosome missegregation seen in H3-G34R and H3-G34K sole copy strains, there was no evidence of increased chromosome loss in the mixed H3 backgrounds for either H3-G34K or H3-G34R (*Figure 6E*). There was however a significant drop in efficiency for H3-G34R mixed copy strains in our plasmid-based HR DNA repair assay (*Figure 6F*).

Finally we asked whether gene expression of H3-G34R and H3-G34K mixed copy strains differed from H3-WT (3xH3). We note that the overall numbers of genes that are differentially expressed in the (3xH3) background are reduced compared with the number of differentially regulated genes in the sole copy strains (54 genes differentially expressed in H3-G34R (3xH3) vs 160 genes in H3-G34R, and 142 genes differentially expressed in H3-G34K (3xH3) vs 223 in H3-G34K when using FDR 5% and FC of 1.5), suggestive that the presence of wild-type H3 partially overrides some transcriptional effects of the mutants. To determine whether the mutations exert dominant effects on expression of particular genes, we looked for overlap between genes that are differentially expressed in the single copy and mixed copy strains. Using the more relaxed criteria of p<0.05, we found that the majority of genes influenced by mutation in the (3xH3) background were also differentially regulated in the sole copy backgrounds (*Figure 6—figure supplement 1B*, *Supplementary file 4*). Manhattan plots

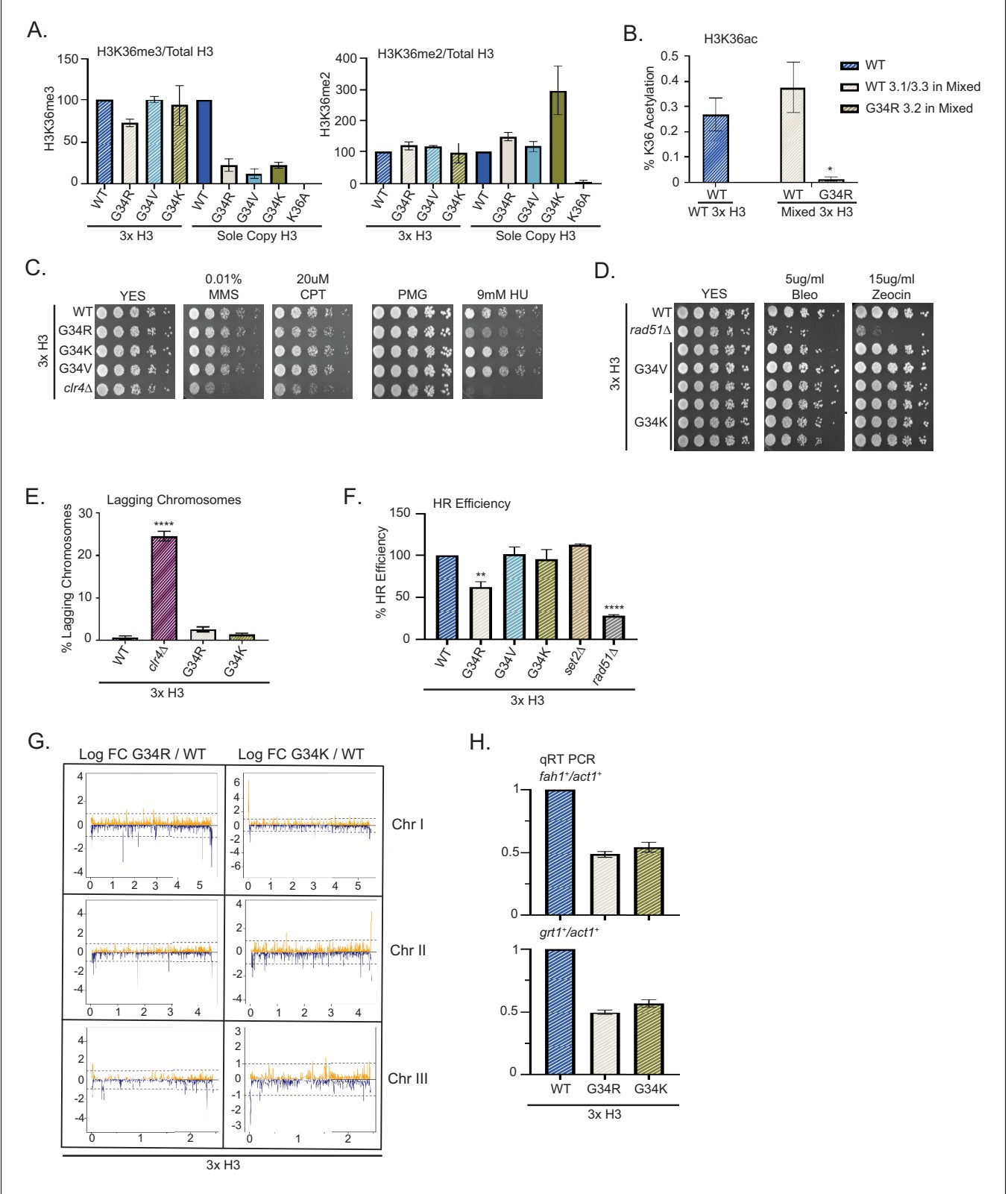

**Figure 6.** Dominance of hydroxyurea (HU) sensitivity, homologous recombination (HR) defects and subtelomeric silencing in strains co-expressing H3-G34R and WT H3. (**A**) Western blot quantification of H3K36me2 and H3K36me3 normalized to total H3 in H3-WT, H3-G34R, H3-G34V, and H3-G34K chromatin extracts in both mixed H3 backgrounds (H3.1/H3.3 WT and H3.2 mutant) and sole copy H3 backgrounds. H3K36me2 and H3K36me3 methylation levels relative to total H3 were calculated from two biological replicates. (**B**) Mass spectrometry-based quantification of acetylation of

*Figure 6 continued on next page*

*Figure 6 continued*

H3K36 on histones purified from H3-WT (3xH3) and H3-G34R (mixed 3xH3 background) from four biological replicates. (C) Fivefold serial dilution growth assays showing the effect of HU, methyl methanesulfonate, and camptothecin on the indicated 3xH3 strains. (D) Serial dilution growth assay showing the bleomycin and zeocin sensitivity of H3-WT, *rad51Δ*, H3-G34V, and H3G34K 3xH3 cells. (E) Frequency of late anaphase cells that show a lagging chromosome in 3xH3 strains: WT, *clr4Δ*, H3-G34R, and H3-G34K. Mean ± SEM from four to eight replicates. **** represents a significance of p<0.0001 with the H3-WT strain. (F) HR assay based on correction of leu1-32 mutation by HR as shown in *Figure 2—figure supplement 1*. Relative HR efficiency is shown as 100% for H3-WT (3xH3), and results are averaged from three independent experiments with error bars representing SEM. ** p<0.005 and **** p<0.0001 reflect significant differences with H3-WT cells. (G) RNA-seq profiles for chromosomes I, II, and III comparing Logfold change ratios for H3-G34R/H3-WT and H3-G34K/H3-WT plotted against chromosome coordinates. Three biological replicates used for each strain, and all strains contain three copies of H3. (H) qRT-PCR validation of *fah1*+ and *grt1*+ expression relative to *adh1*+ expression from two independent biological replicates. Samples were normalized to the WT-H3 strain and shown as the mean ± SEM. Subtelomeric transcripts in H3-G34R and H3G34K cells (3xH3) are reduced when compared with WT.

The online version of this article includes the following figure supplement(s) for figure 6:

**Figure supplement 1.** H3G34R exerts dominance over HU sensitivity, and both G34R and G34K dominantly suppress subtelomeric transcripts.

indicated that subtelomeric domains of chromosomes I and II were repressed in H3-G34R mixed copy strains compared with WT H3 (3xH3), with a similar, but less marked trend also in H3-G34K mixed copy strains (*Figure 6G*). Using real-time PCR analysis of *grt1* and *fah1* transcripts, we validated the repression of subtelomeric genes in both H3-G34R and H3-G34K mixed background strains (*Figure 6H*). Extending this analysis to ask if subtelomeric domains were preferentially repressed in H3-G34R or H3-G34K mixed copy strains (using FDR 5% and FC of 1.5), we found that a high proportion of genes repressed in G34R and G34K mixed copy strains were resident with ST domains (10/30 and 25/69, respectively) compared with only 1/24 and 7/73 of genes activated in G34R and G34K mixed copy strains (Supplemental Table S4). Thus many of the genes that are repressed by the H3-G34R or G34K mutation are situated in ST domains of chromosomes I and II, and the hyper-repression of gene expression within ST domains is dominant since it is evident in the mixed copy background strains.

## Discussion

In this work, we probe the consequence of mutation of Glycine 34 in histone H3 when the mutant provides the only source of H3, and identify phenotypes that dominate when mutants are co-expressed along with wild-type H3. We found a surprising diversity of phenotypes with H3-G34 mutants differentially affecting post-translational modification of nearby K36, DNA damage sensitivity, and transcriptional control.

Given that a single enzyme, Set2, mediates all H3K36 methylation in fission yeast, it is intriguing that we only saw a reduction in H3K36me3, and only in some H3-G34 mutant backgrounds. Set2 may be unable to catalyze the di-methyl to tri-methyl transition on H3K36 when H3G34 is substituted by V, R, or K, but in the absence of a direct assay of in vitro H3K36 methylation activity of purified or recombinant Set2 on the mutant histones, this remains speculative. However, consistent with a defect in tri-methylation, a very recent report shows that activity of the H3K36 tri-methylase SETD2 is blocked on nucleosomes bearing H3.3 G34 mutations, whereas NSD2-mediated di-methylation is unaffected (*Jain et al., 2020*), although in that study, G34R, G34V, and G34W similarly affected tri-methylation, whereas we saw no defect in H3K36me3 in the H3-G34W mutant, or H3-G34M, or G34Q strains. This difference may stem from subtle differences in how H3 binds to the active site of SETD2 compared with Set2, resulting in differing outcomes for mutants such as H3-G34W in the two systems. Insight into Set2 and SETD2's ability to accommodate G34-substituted H3 is limited because the only available structures are of SET2/D2 complexed with its high-affinity ligand K36M H3.3 (*Bilokapic and Halic, 2019*; *Yang et al., 2016*; *Zhang et al., 2017*), which may influence the conformation of the active site. Another possibility is that the differences in K36me3 stem from our analysis of fission yeast H3, which differs from H3.3 at several residues including alanines instead of serines at residues 28 and 31 within the H3 tail (*Matsumoto and Yanagida, 1985*; *Yadav et al., 2017*), which may influence K36 methyltransferase behavior on G34 mutant templates (*Schuhmacher et al., 2020*). However, consistent with previous studies in mammalian cells, we found that co-expression of mutant (H3-G34R, V, or K) with wild-type H3 proteins did not affect

H3K36me3, suggesting that if indeed Set2 activity is altered, loss of Set2-mediated trimethylation of K36 is limited to the mutant H3 tail.

K36 modifications have been linked to DNA damage response. In fission yeast, H3-G34R mutants with reduced K36ac and K36me3 are defective for HR-mediated repair of DNA breaks, whilst H3-G34V mutants, which maintain K36ac but have reduced K36me3, show no defect in HR. Since both mutants (and HR-proficient H3-G34K) have reduced H3K36me3, reduction in K36me3 is not causal of the HR defect. However, like H3-G34R mutants, Gcn5 mutants are reduced for H3K36ac, are HU sensitive, and are also defective for HR (*Pai et al., 2014*); so these phenotypes may be linked. We confirmed that Gcn5 is the major H3K36 acetyltransferase in fission yeast and then showed in vitro that Gcn5 activity was lost specifically at K36 on recombinant G34R H3. This result emphasizes that H3-G34R directly affects Gcn5 function, but meant that we could not bypass the K36 acetylation defect in H3-G34R cells by Gcn5 overexpression to assign specific phenotypes to loss of H3K36ac. We also have not been able to use H3 mutants that mimic acetylated or non-acetylated K36 since they also impact K36 methylation. It therefore remains unclear whether the HR defect and sensitivity to HU seen in G34R cells are caused by loss of H3K36ac. These phenotypes show dominance in mixed copy H3-G34R strains, but there was no global loss of H3K36ac on wild-type H3 purified from mixed copy H3-G34R strains. However, since chromatin is assembled from nucleosomes bearing two copies of H3, one possibility is that even in the presence of excess WT H3, H3K36ac is sufficiently diluted to cause dominant defects in HR and dominant HU sensitivity. In other words, the histone mutants disrupt the local chromatin architecture such that deposition of H3-G34R at *leu1* reduces local levels of K36ac and disrupts HR at that locus. Similar explanations have been suggested for local dominant roles of other histone mutants identified in cancer (*Nacev et al., 2019*). Unfortunately, we cannot readily determine if mixed copy (3xH3) G34R strains show global HR defects using *rad51Δ* in epistasis analyses since the mixed copy H3-G34R strain is only sensitive to HU at concentrations that kill *rad51Δ* in the chronic exposure plating assays. Additionally, we are unable to specifically track the localization of WT and mutant H3 proteins by ChIP since anti-H3G34R antibodies do not recognize fission yeast H3-G34R.

Another possibility would be that the unique phenotypes of H3-G34R mutants are due to the generation of a novel site of arginine methylation within the H3 tail. We have used mass spectroscopy to explore this possibility, but found no evidence for methylation of H3-G34R in fission yeast (*Lowe et al., 2019*).

In contrast to H3-G34R cells, which are not sensitive to γ-IR, both H3-G34V and H3-G34K are sensitive to γ-IR and to IR-mimetics but there was no dominant effect since mixed copy strains were not sensitive to IR mimetics. This may mean that local perturbation of chromatin structure from deposition of the mutant histones is insufficient to yield IR sensitivity in the presence of wild-type H3.

We also found that substitution of G34 with basic residues K or R led to accumulation of Sgo2 at subtelomeric (ST) domains correlating with enhanced silencing of these regions. These mutants also caused chromosome missegregation, but whether these phenotypes are linked is not clear. Cells defective for *gcn5Δ* and *mst2Δ* similarly show repression of ST domains (*Nugent et al., 2010*), and we found that they accumulate Sgo2 at ST domains and, like G34R, accumulate knobs. At ST domains, Sgo2 controls the timing of replication of late replicating origins, and loss of Sgo2 advances replication of some ST origins (*Tashiro et al., 2016*). In mitosis, Sgo2 is located at centromeres where it aids fidelity of chromosome segregation (*Kawashima et al., 2010*). One possibility is that the elevated levels of Sgo2 at ST domains in H3-G34R, H3-G34K, and H3K36 acetyltransferase mutants further delays the timing of late-replicating ST origins, leading to chromosome mis-segregation, as well as replicative stress sensitivity. Alternately, the aberrant accumulation of Sgo2 at ST domains in mitotic H3-G34R cells could indirectly perturb centromere function. This is not due to depletion of Sgo2 from centromeres (as centromeric Sgo2 accumulation is similar between G34R and wild-type cells), but perhaps from mislocalization and/or depletion of a limiting factor that is normally recruited by Sgo2 to centromeres in mitosis, and that in G34R cells is now retargeted preferentially to ST domains. However, we note that silencing of ST domains is maintained in mixed copy strains, whereas chromosome missegregation is corrected.

What is clear from many organisms is that recruitment of Shugoshin proteins relies on Bub1 kinase-mediated phosphorylation of histone H2A on S121 (T120 in mammals) (*Kawashima et al., 2010*; *Kitajima et al., 2005*; *Kitajima et al., 2004*; *Matsuda et al., 2015*; *Tang et al., 2004*; *Tashiro et al., 2016*). In fission yeast, both Bub1 kinase dead and H2A-S121A mutants lose

localization of Sgo2 from centromeres and subtelomeres (*Tashiro et al., 2016*). We previously reported that H3-G34R mutants also alter the kinetics of phosphorylation at residues S128/129 of histone H2A (*Yadav et al., 2017*). Together with our new finding that H3-G34R/K mutants influence the localization of Sgo2, these data suggest the possibility that the G34R and G34K mutant H3s influence signaling through multiple phosphorylation sites clustered on the C-terminus of H2A. This is consistent with structural analyses of nucleosomes showing that the C-terminus of H2A lies close to the site of G34 substitutions on the H3 N-terminal tail (*Du and Briggs, 2010*).

We note that an interesting functional link between H3 and Shugoshin has also been made in budding yeast, where the homolog of Sgo2, SGO1 has been shown to bind histone H3. Although not reliant on G34 or K36 residues of H3, SGO1 binds a 'tension-sensing motif' at residues 42–45 on H3, which is necessary for Sgo1 recruitment to centromeres and for efficient chromosome segregation during mitosis (*Deng and Kuo, 2018*; *Luo et al., 2016*). Intriguingly in this system, the histone acetyltransferase Gcn5 is important for ensuring efficient chromosome segregation when the tension sensing motif is damaged (*Buehl et al., 2018*; *Luo et al., 2016*). It is not clear whether Shugoshin proteins in higher organisms show a similar dependence on N-terminal regions of H3 for their localization and whether H3.3-G34R mutation influences their localization.

Finally we show dominant effects on transcriptional control exerted by the H3-G34R and G34K mutants. In particular, subtelomeric regions of chromosomes I and II are repressed. In keeping with the importance of transcriptional regulation in H3-G34 mutant cells, very recent work from the Lewis lab indicates that H3.3G34 mutants in mammalian cells exert profound transcriptional effects through modulation of Polycomb activity (*Jain et al., 2020*). They found that the decrease in K36me3 on H3.3 G34 mutant histone enhances PRC2 activity to promote elevated H3.3K27me3 in *cis* on the H3.3 tail (*Jain et al., 2020*). They propose that tumorigenesis driven by H3.3G34 mutations derives from enhanced Polycomb-mediated silencing of enhancers of genes that regulate differentiation, whereby cells retain a more stem cell-like fate. Since acetylation at enhancers principally resides on K27 of histone H3.3 (*Martire et al., 2019*), this may explain why the G34 mutations identified in cancer occur solely on H3.3 (*Behjati et al., 2013*; *Schwartzentruber et al., 2012*).

H3.3 only differs from H3.2 by four residues: Ser31 in the tail, and three substitutions within the core of H3 which dictate chaperone selectivity (*Ahmad and Henikoff, 2002*; *Elsässer et al., 2012*; *Liu et al., 2012*). Phosphorylation of Ser31 in H3.3 is important for regulation of cell fate during gastrulation and for enhancer activation (*Martire et al., 2019*; *Sitbon et al., 2020*) and Ser31P influences H3.3 K27 trimethylation and acetylation (*Martire et al., 2019*; *Sitbon et al., 2020*), and so phosphoregulation via S31 may provide an additional mechanistic link as to why the Gly 34 mutations are only found in H3.3.

Although fission yeast provides an exquisite system for analysis of histone mutations, we fully acknowledge the limitation of work in a system that lacks a Polycomb silencing pathway. It is interesting to note however the profound effects of G34R mutation on H3K36ac, the effects of the basic substitutions in the N-terminal tail of H3 on Shugoshin localization and the distinct DNA damage phenotypes of the different H3G34 mutations. It will be interesting to determine the extent to which the phenotypes uncovered in this study are shared in mammalian cancer cells expressing these mutants.

# Materials and methods

Key resources table

| Reagent type (species) or resource | Designation | Source or reference | Identifiers | Additional information |
|---|---|---|---|---|
| Gene | | (*Schizosaccharomyces pombe*) | *hht2*$^+$; H3.2 | Pombase |
| Pombase: SPBC8D2.04 | | | | |
| Strain, strain background (*Escherichia coli*) | BL21(DE3) | Sigma—Aldrich | CMC0016 | |

*Continued on next page*

*Continued*

| Reagent type (species) or resource | Designation | Source or reference | Identifiers | Additional information |
|---|---|---|---|---|
| | | Electrocompetent cells | Antibody | Anti-histone H3 (rabbit polyclonal) |
| Active motif | 39163 | WB. (1:1000) | | |
| Antibody | Anti-histone H3 (rabbit polyclonal) | Abcam | Identifiers | Additional information |
| Ab1791 | WB. (1:1000) | | | |
| Antibody | Antihistone H4 (rabbit monoclonal) | Millipore | 05–858, lot 2020541 | WB. (1:1000) |
| Antibody | Anti-FLAG M2 (mouse monoclonal) | Sigma | F1804 | WB. (1:1000) |
| Antibody | Anti-tubulin (mouse monoclonal) | Gift from Keith Gull, *Woods et al., 1989* | TAT1 | IF (1:100) |
| Antibody | Anti-H3K36me2 (rabbit polyclonal) | Abcam | Ab9049 | WB. (1:1000) |
| Antibody | Anti-H3K36me3 (rabbit polyclonal) | Abcam | Ab9050 | WB. (1:1000) |
| Antibody | Anti-H3K36me3 (rabbit monoclonal) | Cell Signalling Technology | 4909 | WB. (1:1000) |
| Antibody | Anti-H3K36ac (rabbit monoclonal) | Abcam | Ab177179 | WB. (1:1000) |
| Antibody | Anti-H3K36ac (rabbit polyclonal) | Abnova | PAB31320 | WB. (1:1000) |
| Antibody | Anti-H3K36ac (rabbit polyclonal) | Rockland | 600–401-I89 | WB. (1:1000) |
| Antibody | Anti-H3K36ac (rabbit monoclonal) | Thermo Fischer | MA5-24672 | WB. (1:400) |
| Recombinant DNA reagent | leu1+ plasmid | *Yadav et al., 2017*. | JP1050 | Integration plasmid for leu1+ |
| Recombinant DNA reagent | 6xHis-Gcn5-FLAG in pET28a | This paper. Synth by GenScript. | *E. coli* expression vector JP-2587. Codon optimized | For recombinant expression of fission yeast Gcn5. See plasmid DNA in Mat. and Meth. |
| Recombinant DNA reagent | 6xHIS-H3 in pCDF duet | This paper. Synth by GenScript | *E. coli* expression vector JP-2395 | For recombinant expression of fission yeast H3. See plasmid DNA in Mat. And Meth. |

*Continued on next page*

*Continued*

| Reagent type (species) or resource | Designation | Source or reference | Identifiers | Additional information |
|---|---|---|---|---|
| Recombinant DNA reagent | 6xHIS-H3-G34R in pCDF duet | This paper | *E. coli* expression vector JP-2489 | For recombinant expression of fission yeast H3-G34R. See Mat. and Meth. |
| Recombinant DNA reagent | 6xHIS-H3-G34V in pCDF duet | This paper | *E. coli* expression vector JP-2490 | For recombinant expression of fission yeast H3-G34V. See Mat. And Meth. |
| Recombinant DNA reagent | 6xHIS-H3-G34K in pCDF duet | This paper | *E. coli* expression vector JP-2902 | For recombinant expression of fission yeast H3-G34K. see Mat. And Meth. |
| Sequence-based reagent | JPO-183 | *Yadav et al., 2017* | PCR primers for leu1 fragment | GATTTCTGG TCATTTACG TTACTGTA |
| Sequence-based reagent | JPO-3480 | *Yadav et al., 2017* | PCR primers for leu1 fragment | ATCGACACC TTCAACGA TTTC |
| Commercial assay or kit | TruSeq Stranded Total RNA library Prep kit | Illumina | 20020599 | |
| Commercial assay or kit | Ribo-zero Gold rRNA removal kit (yeast) | Illumina | MRZY1324 | Discontinued |
| Commercial assay or kit | RNeasy mini kit | Qiagen | 74104 | |
| Commercial assay or kit | ERCC RNA spike in control | Invitrogen | 4456740 | For normalization of RNAseq |
| Chemical compound, drug | Hydroxyurea | Sigma | H8627 | |
| Chemical compound, drug | Methyl | methanesulfonate | Acros Organics | 66-27-3 |
| Chemical compound, drug | Camptothecin | Sigma | C9911 | |
| Chemical compound, drug | Bleomycin sulfate | Research Products International | B40060 | |
| Chemical compound, drug | Zeocin | Invitrogen | R25001 | |
| Chemical compound, drug | Trypsin | Promega | V5111 | For trypsin digestion of histones |
| Chemical compound, drug | Propionic anhydride | Sigma | 8006080100 | For propionylation of histones |
| Software, algorithm | Xcalibur | Thermo | Version 2.1 | MS peak integration |
| Software, algorithm | GraphPad Prism | Prism | Version 7.03, version 8 | Used at St. Jude, Fox Chase respectively |

*Continued on next page*

*Continued*

| Reagent type (species) or resource | Designation | Source or reference | Identifiers | Additional information |
|---|---|---|---|---|
| Software, algorithm | 3D-SIM reconstruction software | Global Life Sciences Solutions Operations | softWoRx 7.0.0 | Using a homemade optical transfer function |
| Software, algorithm | STAR | *Dobin et al., 2013* | Version 2.5.3a | RNAseq analysis |
| Software, algorithm | Subread R package | *Liao et al., 2019* | | RNAseq analysis |
| Software, algorithm | Limma/ Voom | *Law et al., 2014*; *Ritchie et al., 2015* | | RNAseq analysis |
| Other | Formaldehyde | Polysciences Inc | 18814–20 | For 3D-SIM |
| Other | Zymolyase 100T | Nacalai tesque, Japan | 07665–55 | For 3D-SIM |
| Other | Glycerol | Fuji Film Wako, Japan | 076–00641 | For 3D-SIM |
| Other | Cover glass | Matsunami, Japan | No. 1S | For 3D-SIM |
| Other | 3D-SIM microscope | Global Life Sciences Solutions Operations | DeltaVision\|OMX Version 3 | With a custom 100x UPlanSApo NA1.35 silicone immersion objective lens and a tube lens with a focal length of 75 mm |

## Strain generation

Histone mutant strains were generated as described by *Mellone et al., 2003*. Gene mutagenesis used standard PCR-based procedures, and strains are listed in *Supplementary file 1*. All crosses used random spore analysis with nutritional/drug/temperature selection and PCR verification (including verification of loss of additional H3/H4 genes) and sequencing of H3.2 allele. Two independent clones for each genotype were used in nearly all experiments, and all experiments were performed at least twice.

## Yeast growth media

Fission yeast were maintained on rich (YES), or *pombe* minimal with glutamate (PMG) media with appropriate supplements (*Moreno et al., 1991*). PMG is Edinburgh minimal medium with glutamate.

## Plasmid DNA and recombinant proteins

Codon optimized Gcn5 (*S. pombe*) with 1x FLAG was cloned into pET28a in frame with N-terminal 6x HIS tag by GenScript to generate JP-2587.

Fission yeast histone H3 (and mutant versions [G34R, G34V, G34K]) were codon optimized by use of codon utilization analyzer 2.0 for expression in *E. coli* and the cDNAs were synthesized by IDT and were cloned into pCDF duet (Novagen) in frame with an N-terminal 6x HIS tag for bacterial expression: vectors JP-2395 (WT), JP-2489 (G34R), JP-2490 (G34V), and JP-2902 (G34K).

Recombinant pombe histones were purified from *E. coli* by 'The Histone Source' at Colorado State University following standard procedures for histone purification (*Dyer et al., 2004*).

Recombinant *S. pombe* Gcn5 was purified by the St. Jude protein production core using BL21 (DE3) transformed with JP2587. Four 1 L cultures were grown in LB medium with Kanamycin to an OD600 of about 0.6, then expression was induced with 0.1 mM IPTG and incubation continued for 16 hr at 16˚C. Cell pellets were collected by centrifugation. Cells were suspended in 50 mM Tris.Cl pH 7.9; 500 mM NaCl, 10% glycerol with complete EDTA-free protease inhibitors (Roche) and lysed by two passages through a microfluidizer (Microfluidics Inc). After clarification by centrifugation, the protein was applied to a 1 mL HisTrap column (GE Healthcare) and Gcn5 eluted with a gradient of

imidazole. Proteins were visualized by SDS-PAGE and fractions containing Gcn5 were pooled and dialyzed against 20 mM Tris.Cl, pH 8.0; 200 mM NaCl; 10% glycerol, and 1 mM TCEP.

## Histone purification for mass spectrometric studies

Histones were purified from fission yeast following a previously described purification protocol with some modification (*Sinha et al., 2010*). A 150 mL culture was inoculated to a density of $1.4 \times 10^6$ cells/mL in 4× YES media and grown at 25°C to a density of $3.6 \times 10^7$ cells/mL and harvested by spinning at 3000 rpm for 5 min. Cells were washed with $H_2O$ containing 10 mM sodium butyrate, followed by NIB buffer (250 mM sucrose, 20 mM HEPES pH 7.5, 60 mM KCl, 15 mM NaCl, 5 mM $MgCl_2$, 1 mM $CaCl_2$, 0.8% Triton X-100, 0.5 mM spermine, 2.5 mM spermidine, 10 mM sodium butyrate, 1 mM PMSF, and Sigma yeast protease inhibitor). The pellet was frozen on dry ice and stored at −80°C. For lysis, the pellet was resuspended in 3 mL of NIB and transferred to two 7-mL bead beater tubes along with chilled acid-washed glass beads. Samples were cooled on ice and bead beaten twice for 2 ½ min at max power with 5 min on ice in between. The sample was collected by 'piggy backing' into a 50-mL Oak Ridge tube at 3000 rpm at 4°C in a benchtop centrifuge. The sample was then pelleted by centrifuging at 13,000 rpm for 10 min at 4°C in a Beckman Avanti centrifuge J-30I centrifuge using a JA25.50 rotor. The pellet was washed in 15 mL of NIB before resuspension in 10 mL of 0.4N $H_2SO_4$, sonication for 45 s at max power and incubation for 2 hr on a rotating wheel at 4°C. The sample was pelleted by centrifuging at 13,000 rpm at 4°C for 10 min and the supernatant was transferred to a new tube along with 5 mL of 5% buffer G (5% guanidine HCl and 100 mM potassium phosphate buffer pH 6.8) where the pH was adjusted to 6.8 using 5N KOH. 0.5 mL of Bio-Rex pre-equilibrated in 5% buffer G was added to the sample and incubated at room temperature with rotation overnight. The resin was then washed twice with 20 mL of 5% buffer G and incubated with 3 mL of 40% buffer G (40% guanidine HCl and 100 mM potassium phosphate buffer pH 6.8) for 1 hr at room temperature to elute the bound protein. Buffer exchange and concentration was performed against 5% acetonitrile with 0.1% TFA to a final volume of 150 µL and the sample was stored at −80°C. Protein concentration was measured and 10 µg of sample was electrophoresed and stained with coomassie blue for quality control.

## Acetylation mass spectrometric analyses: UPLC-MS/MS analysis

Acid-extracted histone samples were TCA-precipitated, acetone-washed, and prepared for mass spectrometry analysis as previously described (*Kuo et al., 2014*). A Waters (Milford, MA) Acquity H-class UPLC system coupled to a Thermo (Waltham, MA) TSQ Quantum Access triple-quadrupole (QqQ) mass spectrometer was used to quantify modified histones. Selected reaction monitoring was used to monitor the elution of the acetylated and propionylated tryptic peptides. Transitions were created to study acetylation of pombe H3 wild type and mutants as well as the H4 tails. The detailed transitions for peptides of H3 that vary in sequence from *Xenopus* are reported in *Supplementary file 3*, the transitions for the *Xenopus* peptides and pombe H3G34R have been previously reported (*Kuo et al., 2014*; *Yadav et al., 2017*).

## QqQ MS data analysis

Each acetylated and/or propionylated peak was identified by retention time and specific transitions. The resulting peak integration was conducted using Xcalibur software (version 2.1, Thermo). The fraction of a specific peptide (*Fp*) is calculated as:

$$F_p = I_s / (\Sigma I_p)$$

where $I_s$ is the intensity of a specific peptide state and $I_p$ is the intensity of any state of that peptide.

Data for acetylation analyses of H3-WT, H3-G34R, and *set2Δ* came from nine biological replicates, and six biological replicates for H3-G34V. 1–2 One to two technical replicates from each prep were used, thus data for G34V was obtained from 9 to 12 samples, and for H3-WT, H3-G34R, and *set2Δ* from 15 to 18 samples. Data to examine dominance using histones prepared from 3x H3 strains used six biological replicates for 3xH3: WT, G34R, and *set2Δ*.

## In vitro Gcn5 enzymatic assay

To determine the activity of *S. pombe* Gcn5 on the histone variants (G34V and G34R), recombinant *S. pombe* H3 and mutants were mixed with recombinant Gcn5 in a reaction buffer containing 100 mM ammonium bicarbonate and 50 mM HEPES buffer (pH 7.8) at 37°C. Experiments were performed utilizing excess *S. pombe* histone H3 (15 µM, either wild-type, G34V, or G34R mutant), excess acetyl-CoA (200 µM), in the presence of *S. pombe* Gcn5 (1 µM). Reactions were quenched at time intervals (1, 2.5, 5, 15, and 30 min) with 3x volume TCA and boiled at 95°C for 5 min. Samples were analyzed via UPLC-MS/MS analysis, as described in the section 'Acetylation mass spectrometric analyses: UPLC-MS/MS analysis'. The complete experiment was performed three times.

## Chromosome stability assays

1. Minichromosome loss. The minichromosome (Ch16) (*Niwa et al., 1989*) bears an *ade6-216* allele, which can complement the *ade6-210* allele present within the strain background. Loss of Ch16 causes loss of complementation of function of *ade6$^+$* and accumulation of red pigment when cells are grown on limiting adenine media. Strains containing Ch16 were grown in PMG–Leu (to maintain Ch16) at 32°C to a density of $5 \times 10^6$ cells/mL. Cells were diluted in PMG (no additives) to a final concentration of $5-10 \times 10^3$ cells/mL. Cells were plated onto PMG agar supplemented with amino acids and nucleobases with limiting (10% normal concentration) adenine, incubated at 25°C for 5 days and then transferred to 4°C to let red color develop. Ch16 loss frequencies were calculated by counting half-sectored colonies (and those with >50% red but <100% red) and dividing by the total number of white, white sectored, and less than half red colonies, excluding red colonies from the analysis as they have lost the minichromosome before plating. Results represent data from multiple (2-4) independent cultures of cells, and $5-10 \times 10^3$ colonies were scored for each strain. Samples were assigned a code so identity was masked for this experiment.
2. Lagging chromosome analysis. Chromosome mis-segregation frequencies were obtained as previously described (*Mellone et al., 2003*). Over 400 late anaphase cells were scored for presence of lagging chromosomes for each genotype, using at least three independent cultures of strains. Strain identities were masked for these experiments.

## Western and dot blots

Denatured extracts in 2× SB. Whole cell extracts (WCE) were made as published previously (*Alper et al., 2013*).

### Chromatin fractionation

Fifty milliliters of cells were grown to a density of $6 \times 10^6$ cells/mL at 30°C in YES. Cells were collected by spinning at 3000 rpm for 3 min and washed 3× with 1 mL of ice-cold NIB (described above). After washing, the supernatant was discarded, and the final chromatin pellet was resuspended in 1 mL of NIB. Five hundred microliters of this material was mixed with 500 µL of 4× SDS sample buffer and heated at 98°C for 10 min; 10–20 µL was loaded onto a 12% SDS-PAGE gel for analysis.

### Antibodies used

**Anti-H3** (active motif 39163 lot no. 26311003, Abcam ab1791). **Anti-H4** (millipore 05–858 lot no. 2020541). **Anti-FLAG M2** monoclonal (Sigma F1804), **anti-tubulin** (TAT1 kind gift from Keith Gull) (*Woods et al., 1989*). **Anti-H3K36me2**: Abcam ab9049, **Anti-H3K36me3**: Abcam ab9050, Cell signaling technology 4909. **Anti-H3K36Ac** (Abcam ab177179, rabbit monoclonal), Abnova (PAB31320), Rockland (600–401-I89), and Thermo Fischer (MA5-24672).

Peptide sequences used for assessment of K36 methyl Abs are listed in *Supplementary file 2*.

Slot blots used a twofold serial dilution series with 150 µL of 50, 25, 12.5, 6.25, 3.125, and 1.6 µM peptides spotted on activated PVDF 0.2-micron membrane using a 48-well BioRad Bio-Dot SF apparatus. Spots were air dried for 5 min before staining with Ponceau S stain to verify equal peptide loading. The membrane was blocked in 5% BSA in TBST at RT for 1 hr, incubated with primary Ab for 1 hr at room temperature, washed with TBST, incubated with HRP-conjugated anti-rabbit secondary Ab for 30 min, washed with TBST and then developed with enhanced chemiluminescence

and images captured by LI-COR imaging. Anti-H3 K36 methylation Abs used are listed above in "Antibodies used".

## Western blot quantification

All western blot quantification was done using LI-COR Image Studio software.

## Chromatin immunoprecipitation

ChIP assays were performed similarly to *Alper et al., 2013*, substituting Dynabeads Protein G (Invitrogen 1004D) for the protein G sepharose resin. Set2-FLAG ChIP used Anti-FLAG M2 (Sigma F1804) and monitored Set2 association with *act1⁺* or *clr4⁺* loci as a ratio of signal from input DNA; or Sgo2-FLAG association with subtelomeric regions, or centromeric regions relative to *act1⁺*. Primer sequences used for q-PCR are listed in *Yadav et al., 2017*. Sgo2 ChIP in the *nda3-KM311* background used asynchronous cells grown at 32°C or cells blocked in prometaphase by 8 hr incubation at 18°C.

## Transcript analysis

### RNA-seq studies

Hot phenol extraction was used to prepare the RNA (*Leeds et al., 1991*); 25 mL cultures were grown overnight in PMG complete media at 25°C to a density of $2.5 \times 10^6$ cells/mL. The cells were pelleted by centrifugation and washed in DEPC $H_2O$. The pellet was resuspended in 750 µL TES Buffer (50 mM Tris-HCl pH 7.5, 10 mM EDTA, 100 mM NaCl, 0.5% SDS made in DEPC $H_2O$) along with an equal volume of 5:1 phenol:chloroform pH 4.7 and incubated at 65°C for 1 hr with vortexing for 10 s every 10 min. The samples were then cooled on ice and centrifuged for 5 min at 13,000 rpm. The aqueous phase was transferred to a 2-mL phase lock tube and an additional phenol:chloroform extraction was performed. After centrifugation, the aqueous phase was transferred to a new tube and re-extracted using an equal volume of chloroform. To precipitate the RNA, the aqueous phase was transferred to a 2-mL microcentrifuge tube and three volumes of ice cold ethanol and 1/10$^{th}$ volume 3M NaOAc pH 5.2 were added and the samples were kept at −20°C overnight. The next day, samples were centrifuged at 14,000 rpm and 4°C for 15 min to pellet the RNA. The pellet was washed with ice cold 70% ethanol and air dried for 30 min. A Turbo DNAse (Ambion) reaction was set up with 100 µg of RNA in a 150 uL reaction containing 5 uL of Turbo DNAse. The reaction was incubated at 37°C for 30 min and another 5 µL of Turbo DNAse was added and incubated for an additional 30 min prior to the removal of the DNAse using 50 uL of inactivation beads. An RNeasy Mini kit (Qiagen) was used to further clean up and concentrate the RNA, which was eluted in a final volume of 30 µL of DEPC $H_2O$. Five micrograms of total RNA was diluted to 10 µL, and 2 µL of a 1:20 dilution of ERCC RNA Spike-in mix (Invitrogen) was added to each sample.

RNA-seq was performed by the St. Jude Hartwell Center. Ribosomal RNA was removed from the samples using Ribo-Zero Gold rRNA Removal Kit (Yeast) following manufacturer instructions (Illumina). RNA was quantified using a Quant-iT assay (Life Technology). The quality was checked by 2100 Bioanalyzer RNA 6000 Nano assay (Agilent) or LabChip RNA Pico Sensitivity assay (Perkin Elmer) before library generation. Libraries were prepared from 2 µg of RNA. Libraries were prepared using the TruSeq Strand Total RNA Library Prep Kit, beginning at Elution 2 – Fragment – Prime step immediately preceding cDNA synthesis according to the manufacturer's instructions (Illumina) with the following modifications the 94°C Elution 2 – Fragment – Prime incubation was reduced to 5 min and the PCR was reduced to 11 cycles. Libraries were quantified using the Quant-iT PicoGreen dsDNA assay (Life Technologies) or Kapa Library Quantification kit (Kapa Biosystems). One hundred cycle paired end sequencing was performed on an Illumina HiSeq 4000 (single copy strains) or on an Illumina Novaseq 6000 (triple copy strains). Three biological replicates were used for each strain analyzed. The total RNA was sequenced using stranded protocol with $2 \times 100$ bp setting. The paired end reads were mapped to *S. pombe* (v2.29) genome using STAR (v2.5.3a) (*Dobin et al., 2013*). Reads counts for each gene were counted using the subread R package (*Liao et al., 2019*). Raw counts were TMM normalized, and differentially expressed genes were analyzed using linear model of the mean-variance trend using the limma and voom packages in R. LogFC values were produced using the limma/voom packages (*Law et al., 2014*; *Ritchie et al., 2015*). External RNA spike-ins were analyzed to confirm that there were no changes in global RNA expression between the strain

sets under analysis. Processed data are included in Supplemental Table S4. RNA-seq data has been submitted to GEO. Accession no. GSE # GSE162572.

## Real-time q-PCR

Real-time q-PCR was performed on random primed cDNA generated from two independent RNA preps for each strain as previously described (*Debeauchamp et al., 2008*; *Partridge et al., 2007*) using primers that had been tested for linear amplification parameters, and working within the Ct range of linear amplification, and using an Eppendorf Mastercycler RealPlex$^2$ machine. Transcript levels were normalized to *adh1*$^+$ or *act1*$^+$ transcripts.

## Observation of knobs

Knobs were observed as described previously (*Matsuda et al., 2015*). The *S. pombe* cells were cultured in liquid minimum medium with supplements (EMM2 5S) at 26°C with shaking to the early logarithmic phase. Cells were pelleted by gentle centrifugation, and chemically fixed by re-suspending in a buffer containing 4% formaldehyde (Polysciences, Inc, Warrington, PA), 80 mM HEPES-K, 35 mM HEPES-Na, 2 mM EDTA, 0.5 mM EGTA, 0.5 mM spermidine, 0.2 mM spermine, and 15 mM 2-mercaptoethanol, pH 7.0. After fixation for 10 min at room temperature, cells were washed with PEMS (100 mM PIPES, 1 mM EGTA, 1 mM MgSO$_4$, 1.2 M sorbitol pH 6.9) three times, then digested with 0.6 mg/mL zymolyase 100T (Seikagaku Biobusiness, Tokyo, Japan) in PEMS at 36°C for 5 min. Next, cells were treated with 0.1% Triton X-100 in PEMS for 5 min and washed three times thereafter with PEMS. Cells were incubated with 0.2 μg/mL DAPI in PEMS for 10 min, and then cells were resuspended with nPG-Glycerol (100% glycerol for absorption metric-analysis [Wako, Osaka, Japan] with 4% *n*-propyl gallate, pH 7.0) diluted with 1:1 with PEM (PEMS without sorbitol) and mounted on a clean 18 × 18 mm coverslip (No. 1S, Matsunami, Osaka, Japan). The slides were observed with 3D-SIM using a DeltaVision|OMX microscope version two or SR (Global Life Sciences Solutions Operations) equipped with a 100x UPlanSApo NA1.35 silicone immersion objective or 60X UPlanApo NA1.42 oil immersion objective lens (Olympus, Tokyo, Japan). Reconstruction of 3D-SIM was performed by softWoRx (Global Life Sciences Solutions Operations) with Wiener filter constants of 0.003 using a homemade optical transfer function. Conspicuously condensed, DAPI-stained bodies were counted as knobs by visually inspecting each optical section of 3D-SIM images. Experiments were performed with three biological replicates for each strain, and a cumulative total of 189–293 nuclei were examined for each strain. Error bars represent the SEM.

## Serial dilution analyses

Fivefold serial dilution assays were performed using exponentially growing cells and were spotted on agar plates using a plate replicator with $2 \times 10^4$ cells in spot 1. Plates were incubated at indicated temperatures. For chronic exposure assay, a fivefold serial dilution was spotted onto PMG complete agar plates ± HU, on YES agar plates with DMSO or DMSO and CPT, and on YES agar plates ± MMS. Plates were photographed after 4–5 days incubation at 30°C (CPT and MMS), or 6–7 days at 25°C (HU). All experiments were repeated at least twice.

## γ-IR treatment

Cells at a density of $5 \times 10^6$ cells/mL were irradiated using a Cobalt source, and 100 μL samples were taken following increments of 200 Gy exposure (*Pai et al., 2014*). Cells were plated on six YES plates for each assay condition, and colonies scored following incubation at 32°C for 4 days. The experiment was performed twice (four biological replicates) to obtain average viability of treated versus untreated cells, and error bars represent the SEM. Strain identities were masked.

## HR assay

We transformed *leu1-32* mutant cells (that bear a single nucleotide mutation in *leu1* that renders cells auxotrophic for leucine) with a fragment of wild type *leu1*$^+$ (amplified using JPO 183/JPO 3480) and scored *leu1*$^+$ transformants that can only arise by HR of the wild-type *leu1* fragment into the mutant allele (*Yadav et al., 2017*). The rates of HR were normalized by calculating transformation efficiencies of the different strains using *leu1*$^+$ plasmid DNA (JP1050). The experiment was performed 3–4 times, with error bars representing SEM. Strain identities were masked.

## Statistical analyses

Statistical analyses were performed in GraphPad Prism version 7.03, using ordinary one-way ANOVA or repeated measures ANOVA and Dunnett's multiple comparisons test.

## Acknowledgements

We thank G Zambetti for facilitating extension of RKY's studies; L Doorley and T Hall for help with preliminary assays; and R Allshire, J Tyler, S White, S Dent, and M O'Connell for useful discussions. JFP thanks L Hendershot and T Lawrence for support. We thank F Bachand, S Jia, E Hidalgo, S Forsburg, A Annunziato, M O'Connell, B Strahl, R Allshire, and J Kanoh for gifts of strains. We thank St. Jude protein production core for purification of Gcn5, St. Jude Hartwell Center staff for RNA-seq library preparation, DNA sequencing and peptide synthesis, and J Riggs (ARC) for help with the cesium source.

## Additional information

### Funding

| Funder | Grant reference number | Author |
| --- | --- | --- |
| St. Baldrick's Foundation | Research grant with generous support from the Henry Cermak fund for Pediatric Cancer Research | Janet F Partridge |
| National Cancer Institute | Cancer Center support grant (NCI CCSG 2 P30 CA21765) | Rajesh K Yadav Janet F Partridge |
| American Lebanese Syrian Associated Charities | | Brandon R Lowe Rajesh K Yadav Patrick Schreiner Alfonso G Fernandez David Finkelstein Margaret Campbell Satish Kallappagoudar Carolyn M Jablonowski Janet F Partridge |
| National Institutes of Health | NIH GM102503 | Andrew J Andrews |
| Fox Chase Cancer Center | Board of Associates Fellowship | Ryan A Henry |
| Japan Society for the Promotion of Science | Kakheni grant JP19H03202 | Atsushi Matsuda |
| Japan Society for the Promotion of Science | Kakheni grants JP18H05533 | Yasushi Hiraoka |
| Japan Society for the Promotion of Science | JP20H05894 | Atsushi Matsuda |
| Japan Society for the Promotion of Science | JP20H00454 | Yasushi Hiraoka |

The funders had no role in study design, data collection and interpretation, or the decision to submit the work for publication.

### Author contributions

Brandon R Lowe, Conceptualization, Validation, Investigation, Visualization, Methodology, Writing - original draft, Writing - review and editing; Rajesh K Yadav, Investigation, Methodology, Writing - review and editing; Ryan A Henry, Investigation, Visualization, Methodology, Writing - review and editing; Patrick Schreiner, Software, Formal analysis, Visualization, Writing - review and editing; Atsushi Matsuda, Funding acquisition, Investigation, Visualization, Methodology, Writing - review and editing; Alfonso G Fernandez, Satish Kallappagoudar, Investigation, Methodology; David Finkelstein, Software, Formal analysis; Margaret Campbell, Carolyn M Jablonowski, Investigation; Andrew

J Andrews, Yasushi Hiraoka, Resources, Supervision, Funding acquisition, Validation; Janet F Partridge, Conceptualization, Supervision, Funding acquisition, Validation, Investigation, Visualization, Methodology, Writing - original draft, Project administration, Writing - review and editing

## Author ORCIDs
Patrick Schreiner (iD) http://orcid.org/0000-0001-5391-2642
Atsushi Matsuda (iD) http://orcid.org/0000-0003-0510-213X
Yasushi Hiraoka (iD) http://orcid.org/0000-0001-9407-8228
Janet F Partridge (iD) https://orcid.org/0000-0003-1102-6305

## Decision letter and Author response
Decision letter https://doi.org/10.7554/eLife.65369.sa1
Author response https://doi.org/10.7554/eLife.65369.sa2

# Additional files

## Supplementary files

• Supplementary file 1. Strain list.

• Supplementary file 2. Peptides used for antibody characterization and for mass spectrometry calibration.

• Supplementary file 3. Detection parameters of unique tryptic peptides from *S. pombe* H3.

• Supplementary file 4. RNA-seq data.

• Transparent reporting form

## Data availability

RNAseq data have been deposited in GEO under accession code GSE162572.

The following dataset was generated:

| Author(s) | Year | Dataset title | Dataset URL | Database and Identifier |
|---|---|---|---|---|
| Lowe BR, Yadav RK, Henry RA, Schreiner P, Matsuda A, Fernandez AG, Finkelstein D, Campbell M, Kallappagoudar S, Jablonowski CM, Andrews AJ, Hiraoka Y, Partridge JF | 2021 | Surprising phenotypic diversity of cancer-associated mutations at Gly 34 in the histone H3 tail | https://www.ncbi.nlm.nih.gov/geo/query/acc.cgi?acc=GSE162572 | NCBI Gene Expression Omnibus, GSE162572 |

The following previously published datasets were used:

| Author(s) | Year | Dataset title | Dataset URL | Database and Identifier |
|---|---|---|---|---|
| Nugent R, Johnsson A, Fleharty B, Gogol M, Xue-Franzén Y, Seidel C, Wright A, Forsburg SL | 2010 | Expression profiling of *S. pombe* acetyltransferase mutants identifies redundant pathways of gene regulation | https://www.ncbi.nlm.nih.gov/geo/query/acc.cgi?acc=GSE17298 | NCBI Gene Expression Omnibus, GSE17298 |
| Nugent R, Johnsson A, Fleharty B, Gogol M, Xue-Franzén Y, Seidel C, Wright A, Forsburg SL | 2009 | *S. pombe* acetyltransferase mutants identifies redundant pathways of gene regulation, dual-channel dataset | https://www.ncbi.nlm.nih.gov/geo/query/acc.cgi?acc=GSE17259 | NCBI Gene Expression Omnibus, GSE17259 |
| Nugent R, | 2009 | *S. pombe* acetyltransferase mutants | https://www.ncbi.nlm. | NCBI Gene |

Johnsson A, Fleharty B, Gogol M, Xue-Franzén Y, Seidel C, Wright A, Forsburg SL | identifies redundant pathways of gene regulation, Affymetrix dataset | nih.gov/geo/query/acc. cgi?acc=GSE17262 | Expression Omnibus, GSE17262

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
