## [Decision Letter]

**Acceptance summary:**

This article uses yeast to examine the phenotypes of Gly 34 mutations in the histone H3 amino terminal tail. These mutations are found in human cancers. Surprisingly mutation of Gly 34 to different amino acids gives rise to different phenotypes. This suggest that these different mutations are likely to have different effects in cancer cells as well.

**Decision letter after peer review:**

Thank you for submitting your article "Surprising Phenotypic Diversity of Cancer-associated mutations of Gly 34 in the Histone H3 tail" for consideration by *eLife*. Your article has been reviewed by three peer reviewers, including Jerry L Workman as the Reviewing Editor and Reviewer #1, and the evaluation has been overseen by Jessica Tyler as the Senior Editor.

Summary:

This is an exhaustive study of different phenotypes associated with Histone H3-G34 mutations in a fission yeast model. Because mutations at this site occur in certain human cancers, teasing apart their different phenotypes in a model system helps to understand their potential effects in pathology. The phenotypes vary widely, suggesting a key role for this residue in a variety of genome maintenance functions.

Essential revisions:

This is overall a technically very well done paper with a variety of methods to examine different mutations in H3-G34. The strength is the consistent approach applied to numerous mutations. However, as there is no single response, it's rather descriptive overall. We have no major concerns about the data, but feel that the conclusions need to be tempered in two areas where the assays were not direct.

1) In the absence of NHEJ repair assays it needs to be noted that conclusions about NHEJ proficiency based on drug sensitivity are indirect.

2) The authors imply that the H3G34 mutants affect the activity of the Set2 H3K36 methyltransferase. In the absence of an in vitro H3K36 methylation assay on the mutant histones with recombinant or affinity purified Set2 the authors need to note that this conclusion is speculative as they have not measured it directly.

---

## [Author Response]

Essential revisions:This is overall a technically very well done paper with a variety of methods to examine different mutations in H3-G34. The strength is the consistent approach applied to numerous mutations. However, as there is no single response, it's rather descriptive overall. We have no major concerns about the data, but feel that the conclusions need to be tempered in two areas where the assays were not direct.1) In the absence of NHEJ repair assays it needs to be noted that conclusions about NHEJ proficiency based on drug sensitivity are indirect.2) The authors imply that the H3G34 mutants affect the activity of the Set2 H3K36 methyltransferase. In the absence of an in vitro H3K36 methylation assay on the mutant histones with recombinant or affinity purified Set2 the authors need to note that this conclusion is speculative as they have not measured it directly.

As requested, we have tempered the language in respect to the conclusions drawn about NHEJ proficiency, and about Set2 activity (Results and Discussion respectively). These modifications have been tracked along with inclusion of a new grant funding Atsushi Matsuda, an addition to the Materials and methods section and inclusion of the key resource table. I hope it is acceptable to keep separate tables for the fission yeast strain list (Supplementary file 1) and list of peptides used (Supplementary file 2).